# Conserved structural elements specialize ATAD1 as a membrane protein extraction machine

Lan Wang[1,2]*[†], Hannah Toutkoushian[1,2†‡], Vladislav Belyy[1,2], Claire Y Kokontis[1], Peter Walter[1,2]*[‡]

[1]Department of Biochemistry and Biophysics, University of California at San Francisco, San Francisco, United States; [2]Howard Hughes Medical Institute, University of California at San Francisco, San Francisco, United States

*For correspondence:
lan@walterlab.ucsf.edu (LW);
pwalter@altoslabs.com (PW)

[†]These authors contributed equally to this work

Present address: [‡]Altos Labs, San Francisco, United States

Competing interest: The authors declare that no competing interests exist.

**Abstract** The mitochondrial AAA (ATPase Associated with diverse cellular Activities) protein ATAD1 (in humans; Msp1 in yeast) removes mislocalized membrane proteins, as well as stuck import substrates from the mitochondrial outer membrane, facilitating their re-insertion into their cognate organelles and maintaining mitochondria's protein import capacity. In doing so, it helps to maintain proteostasis in mitochondria. How ATAD1 tackles the energetic challenge to extract hydrophobic membrane proteins from the lipid bilayer and what structural features adapt ATAD1 for its particular function has remained a mystery. Previously, we determined the structure of Msp1 in complex with a peptide substrate (Wang et al., 2020). The structure showed that Msp1's mechanism follows the general principle established for AAA proteins while adopting several structural features that specialize it for its function. Among these features in Msp1 was the utilization of multiple aromatic amino acids to firmly grip the substrate in the central pore. However, it was not clear whether the aromatic nature of these amino acids were required, or if they could be functionally replaced by aliphatic amino acids. In this work, we determined the cryo-EM structures of the human ATAD1 in complex with a peptide substrate at near atomic resolution. The structures show that phylogenetically conserved structural elements adapt ATAD1 for its function while generally adopting a conserved mechanism shared by many AAA proteins. We developed a microscopy-based assay reporting on protein mislocalization, with which we directly assessed ATAD1's activity in live cells and showed that both aromatic amino acids in pore-loop 1 are required for ATAD1's function and cannot be substituted by aliphatic amino acids. A short α-helix at the C-terminus strongly facilitates ATAD1's oligomerization, a structural feature that distinguishes ATAD1 from its closely related proteins.

## Editor's evaluation

This study reports the cryo-EM structure of human ATAD1 (Msp1 in yeast), a AAA protein responsible for the extraction of mistargeted tail-anchored (TA) proteins from the mitochondrial outer membrane. The structure helps to understand the effects of disease-linked mutations on ATAD1/Msp1 activity.

## Introduction

Mitochondria serve a multitude of functions, including ATP production, metabolism, and proteostasis that all require import of proteins from the cytosol (*Pfanner et al., 2019*). To ensure proper function, multiple mechanisms facilitate faithful protein targeting and efficient protein import into the organelle. One of these mechanisms is carried out by a protein on the mitochondrial outer membrane (MOM)

named ATAD1 (in humans; Msp1 in yeast). ATAD1 has important roles in various biological contexts (*Fresenius and Wohlever, 2019*; *Wang and Walter, 2020*), including clearing out mistargeted proteins from the mitochondria (*Chen et al., 2014*; *Nuebel et al., 2021*; *Okreglak and Walter, 2014*), extracting mitochondrial precursor proteins stuck in the protein translocase channel during protein import overload (*Weidberg and Amon, 2018*), and mediating apoptosis by removing a BCL-family member protein (*Winter et al., 2021*). In addition, ATAD1 has also been implicated in the regulation of synaptic plasticity by mediating the endocytosis of neurotransmitter receptors from post-synaptic membranes (*Wang and Walter, 2020*; *Zhang et al., 2011*). Among these, the removal of mistargeted tail-anchored (TA) proteins from the MOM is ATAD1's best understood function to date.

TA proteins are integral membrane proteins that are embedded in the membrane by a hydrophobic stretch at the extreme C-terminus. They are targeted to organellar membranes post-translationally. The endoplasmic reticulum (ER)-targeted TA proteins are delivered to the ER membrane by the TRC proteins (Get proteins in yeast) (*Favaloro et al., 2008*; *Schuldiner et al., 2008*; *Stefanovic and Hegde, 2007*). From there, they either stay on the ER membrane or travel to the peroxisome or other membranes along the secretory pathway. By contrast, TA proteins on the MOM either insert spontaneously (*Chio et al., 2017*) or are actively inserted by the mitochondrial import machinery (*Doan et al., 2020*). The partitioning of the ER- and the mitochondria-targeted TA proteins relies on different biophysical properties of their targeting signals (consisting of the transmembrane domain and the short segment C-terminal to it) (*Chio et al., 2017*). Due to a number of similarities between the ER- and mitochondria-targeted TA proteins, mistargeting happens in both directions. Recently, our lab and the Rutter lab independently discovered that Msp1/ATAD1 could recognize the mislocalized TA proteins and extract them from mitochondria (*Chen et al., 2014*; *Okreglak and Walter, 2014*). The extracted proteins are then correctly inserted into the ER membrane, from which they travel to their cognate organelle or become degraded by the proteasome (*Dederer et al., 2019*; *Matsumoto et al., 2019*).

ATAD1 is a member of a large family of proteins called the AAA (A̲TPase A̲ssociated with diverse cellular A̲ctivities) proteins. It is strictly conserved from yeast to humans. Previously, we determined a series of cryo-EM structures of the cytosolic domain of the *Chaetomium thermophilum* (*C.t.*) Msp1 in complex with a peptide substrate (*Wang et al., 2020*). The structures revealed that Msp1 follows the general principle established for many AAA proteins (*Gates et al., 2017*; *Monroe et al., 2017*; *de la Peña et al., 2018*; *Puchades et al., 2020*; *Puchades et al., 2017*; *Zehr et al., 2017*): six Msp1 subunits form a helical hexamer resembling a right-handed lock-washer that surrounds the substrate in a hydrophobic central pore. Elements at the intersubunit interface couple ATP hydrolysis with step-wise subunit translocation to unfold the substrate peptide in its central pore. The structures also revealed elements in Msp1 that are adapted for its function of removing membrane proteins, one of which is an unusually hydrophobic central pore. Whereas most AAA proteins extend only one short loop (pore-loop 1) containing a conserved aromatic amino acid to directly contact the substrate, Msp1 utilizes a total of three aromatic amino acids within two short loops to contact the substrate. We previously proposed that these aromatic amino acids enhance the bulkiness and hydrophobicity of the pore-loops, giving Msp1 a firm grip on the substrate to prevent it from backsliding, a feature that may be important for pulling hydrophobic membrane proteins out of the lipid bilayer.

To test the importance of the additional aromatic amino acids, we previously mutated the second (Y167 in *Saccharomyces cerevisiae* (*S.c.*), corresponding to Y188 in *C.t.*) and the third (H206 in *S.c.*, corresponding to H227 in *C.t.*) aromatic amino acids to aliphatic or polar amino acids and measured their effects on yeast growth. Surprisingly, while the Y167A mutation impacted Msp1's activity, the Y167V or H206A mutation did not have a significant impact on Msp1's function. We hypothesized that this was perhaps because the yeast growth assay provides an indirect readout (cell death) of Msp1's function. In other words, deficiencies in the protein's activity could be masked by compensatory pathways promoting cell survival. Therefore, in this study, we aimed to establish an assay that would allow the direct visualization of protein mislocalization.

In addition to addressing these remaining questions concerning the pore-loops, we also explored unique structural elements in ATAD1 that are not observed in its closely related family members. Within the large AAA protein family, ATAD1 belongs to the 'meiotic clade' (AAA$_{MC}$). Members of this clade include microtubule severing proteins such as katanin and spastin, as well as the protein that disassembles the ESCRT peptides, Vps4 (*Frickey and Lupas, 2004*). One of the unique structural

elements shared by these family members is a C-terminal extension of the classic AAA protein fold, helix α12. α12 is important in hexamer assembly and protein function for AAA$_{MC}$ proteins (*Sandate et al., 2019*; *Vajjhala et al., 2008*). The secondary structure prediction of ATAD1, however, shows the lack of α12, and instead predicts a longer α11 (the alpha helix proceeding it, *Figure 1—figure supplement 1*). Of note, there are several disease-relevant mutations clustered around α11, making it particularly important to understand this region in molecular detail: R9H, D221H, and E290K are found in schizophrenia patients (*Umanah et al., 2017*); H357Rfs*15 (a frame shift mutation resulting in a 10 amino acid extension at the C-terminus) is found in a family of patients with encephalopathy (*Piard et al., 2018*); and E267stop results in a truncated protein that is missing a large portion of the C-terminal domain and is found in a family of patients suffering from hypertonia, seizure, and death (*Ahrens-Nicklas et al., 2017*). In the *C.t.* Msp1 structure, we did not observe density for α11, which could be due to degradation or lack of rigid secondary structure, leaving the structure and function of α11 unresolved. To address these questions, we determined the structure of ATAD1 and tested the individual functional contributions of salient structural features in a microscopy-based mislocalization assay.

## Results

To obtain a homogenous sample for structural analysis, we expressed human ATAD1 lacking the first 40 amino acids (consisting of the transmembrane helix) with an N-terminal His-tag and bearing a commonly used 'Walker B' mutation that inactivates ATP hydrolysis but retains ATP binding (Δ40-ATAD1$^{E193Q}$). Δ40-ATAD1$^{E193Q}$ formed oligomers as assessed by size exclusion chromatography (*Figure 1—figure supplement 2*). We took the fraction corresponding to hexamers and incubated it with ATP before preparing samples for cryo-EM imaging.

3D classification of the particles resulted in two distinct hexameric structures. In the first structure, the six ATAD1 subunits (M1–M6) are arranged in a right-handed spiral with an open seam between the top and the bottom subunits (45,003 particles analyzed, *Figure 1—figure supplement 3B*, *Table 1*), an arrangement resembling the 'open state' structure of Msp1 (*Wang et al., 2020*), as well as other related AAA proteins (*Cooney et al., 2019*; *Han et al., 2020*; *Su et al., 2017*; *Sun et al., 2017*; *Twomey et al., 2019*; *Zehr et al., 2017*; *Figure 1A*). Refinement of this structure resulted in a map with an average resolution of 3.7 Å. In the second structure, the subunits are arranged in a similar fashion as the first, with the exception of M6, which is in transition to convert into M1 as part of the AAA protein functional ATPase cycle and now bridges M1 and M5, closing the hexameric ring (96,577 particles analyzed). This structure closely resembles the 'closed state' structure observed in Msp1 (*Figure 1B*). Refinement of this structure yielded a map with an average resolution of 3.2 Å with most of the side chain densities clearly visible, which allowed us to build an atomic model. Noticeably, the density of the M6 subunit in the 'closed state' structure was less well defined than the rest of the complex (local resolution ranging from 4 to 6 Å, *Figure 1—figure supplement 3C*), indicating a mixture of different states. A similar dynamic nature of this subunit was also observed in the Msp1 structure (*Wang et al., 2020*) but the density was too poorly resolved to be interpreted. By contrast, for ATAD1, we could identify secondary structure elements clearly, which enabled us to build a poly-alanine model for this subunit. In doing so, we were able to analyze the interactions between M6 and neighboring subunits in the closed state, which was not possible for Msp1.

### Hinge motion between the large and the small AAA domains accompanies subunit translocation

The overall architecture of ATAD1 closely resembles that of Msp1, pointing to significant structure/function conservation throughout eukaryotic evolution. Six ATAD1 subunits rotate and translocate progressively to form a hexameric spiral assembly. Each ATAD1 subunit consists of two subdomains, a large subdomain followed by a small subdomain (*Figure 1C*), and the nucleotide is bound at the interface between the two. In the 'open state', all six subunits adopt similar overall structures. To initiate the subunit translocation, the M6 subunit loses its bound nucleotide, and translocates upward to bridge the gap with M1 (*Figure 1—figure supplement 4A*). Meanwhile, the small subdomain of M5 rotates toward its large subdomain, allowing it to establish contacts with the small subdomain of M6 in the 'closed state' structure (*Figure 1—figure supplement 4B*). In the next step of translocation, M6

**Table 1.** Cryo-EM data analysis.

| Structure | Δ40-ATAD1$^{E193Q}$ (closed)<br>PDB ID: 7UPR<br>EMD-26674 | Δ40-ATAD1$^{E193Q}$ (open)<br>PDB ID: 7UPT<br>EMD-26675 |
|---|---|---|
| **Data collection** | | |
| Microscope | Titan Krios | |
| Voltage (keV) | 300 | |
| Nominal magnification | 105,000× | |
| Exposure navigation | Image shift | |
| Electron dose (e⁻/Å²) | 67 | |
| Dose rate (e⁻/pixel/s) | 15 | |
| Detector | K3 summit | |
| Pixel size (Å) | 0.832 | |
| Defocus range (μm) | 0.6–2.0 | |
| Micrographs | 6937 | |
| **Reconstruction** | | |
| Total extracted particles (no.) | 478,463 | |
| Final particles (no.) | 96,577 | 45,003 |
| Symmetry imposed | C1 | C1 |
| FSC average resolution, masked (Å) | 3.2 | 3.5 |
| FSC average resolution, unmasked (Å) | 4.1 | 4.6 |
| Applied B-factor (Å) | 121.9 | 119.2 |
| Reconstruction package | Cryosparc v2.15 and Relion 3.0.8 | |
| **Refinement** | | |
| Protein residues | 1678 | 1694 |
| Ligands | 19 | 19 |
| RMSD bond lengths (Å) | 0.002 | 0.003 |
| RMSD bond angles (°) | 0.571 | 0.802 |
| Ramachandran outliers (%) | 0.00 | 0.06 |
| Ramachandran allowed (%) | 7.10 | 7.34 |
| Ramachandran favored (%) | 92.90 | 92.60 |
| Poor rotamers (%) | 4.18 | 11.28 |
| CaBLAM outliers (%) | 5.69 | 5.39 |
| Molprobity score | 2.33 | 2.79 |
| Clash score (all atoms) | 7.82 | 10.14 |
| B-factors (protein) | 128.89 | 172.35 |
| B-factors (ligands) | 106.10 | 162.46 |
| EMRinger score | 1.45 | 1.19 |
| Refinement package | Phenix 1.17.1-3660-000 | |

continues to move upward, assuming the M1 position (top position) in the open spiral, detaching fully from M5, which now becomes the new M6, occupying the bottom position. Consequently, the angle between the two subdomains of M5 widens, and the subunit resumes its original conformation as it reaches the M6 position (*Figure 1—figure supplement 4B, C*). Similar hinge motions between the two subdomains were observed previously: the crystal structure of the monomeric *S.c.* Msp1 showed

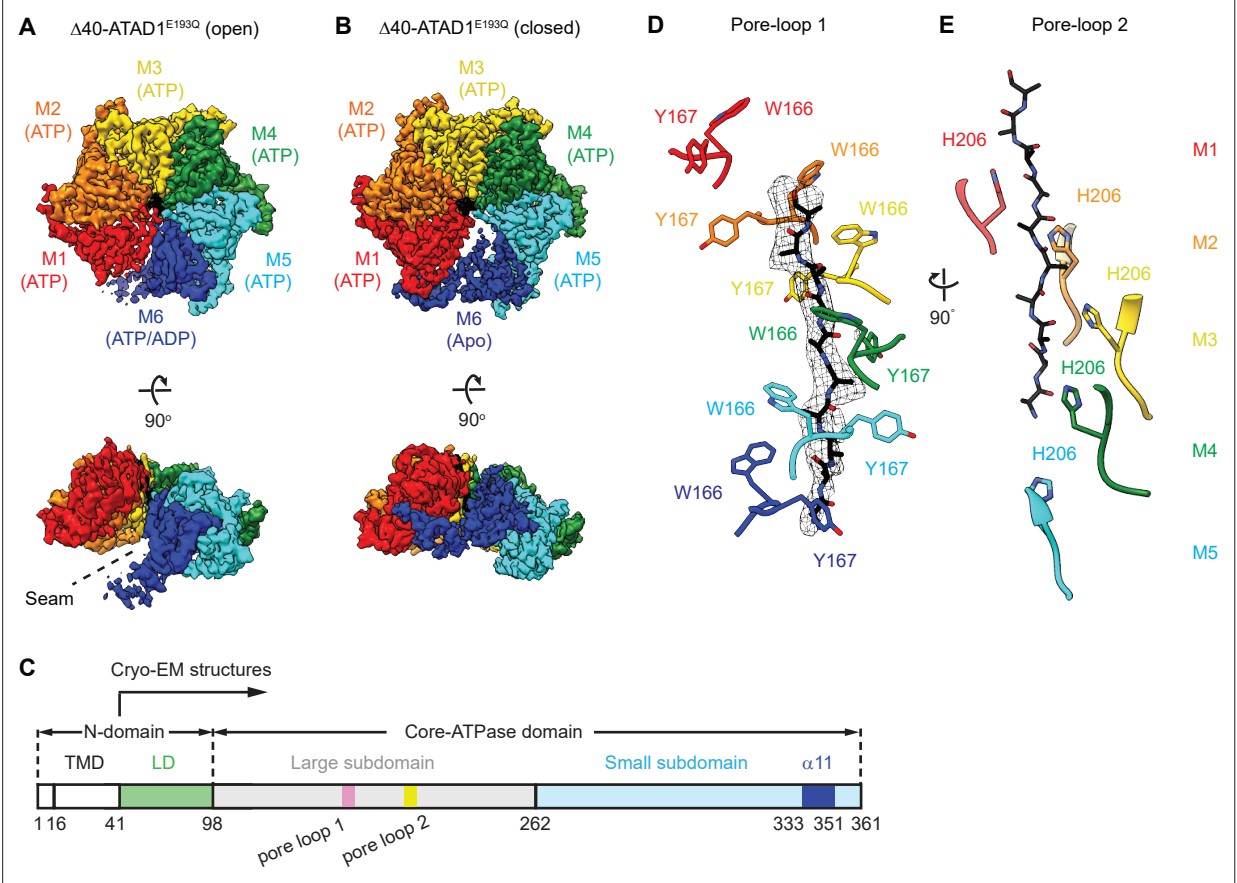

**Figure 1.** Architecture of the Δ40-ATAD1[E193Q]-substrate complexes. (**A and B**) Final reconstructions of Δ40-ATAD1[E193Q] (open) and Δ40-ATAD1[E193Q] (closed) complexes shown in top and side views. Each subunit (M1–M6) is assigned a distinct color, and the substrate is shown in black. The spiral seam of the open conformation (panel A) is denoted with dashed lines. In the top views, the membrane-facing side of ATAD1 is shown. (**C**) Schematic of individual domains and structural elements of human ATAD1. (**D**) Pore-loops 1 (showing those of the closed conformation) form a staircase around the substrate. The peptide density is shown in black mesh. (**E**) Pore-loops 2 form a second staircase below pore-loops 1. H206s directly contact the peptide backbone carbonyls.

The online version of this article includes the following figure supplement(s) for figure 1:

**Figure supplement 1.** Sequence alignment of AAA[MC] proteins.

**Figure supplement 2.** Size exclusion chromatography (SEC) trace of Δ40-ATAD1[E193Q].

**Figure supplement 3.** Cryo-EM processing of the Δ40-ATAD1[E193Q]-substrate complexes.

**Figure supplement 4.** Hinge motion between the large and the small subdomain accompanies subunit movements.

a near 180-degree flip between the small and the large AAA domains and complete disruption of the nucleotide-binding pocket (**Wohlever et al., 2017**). It remained unclear, however, whether the flip resulted from crystal packing forces or reflected a functionally relevant conformational change. The hinge motion in ATAD1 observed here strongly argues that the two domains are connected by a flexible linker and undergo significant rotations relative to each other in a nucleotide- and subunit position-dependent manner.

## A unique α-helix at the C-terminus mediates intersubunit interactions

While examining the interactions of the mobile subunit (M6) in the closed conformation, we noticed that it is held in place by contacts on both sides: on the side of M5, substantial contacts exist between the large and the small subdomains of the two subunits (**Figure 1—figure supplement 4D**). By contrast, on the side of M1, M6 is held in place by two contact points. One structural element stood out as it was not observed previously in the *C.t.* Msp1 structure (**Wang et al., 2020**): an α-helix at the extreme C-terminus of M6 points toward the core β sheet of the large subdomain of M1. We next

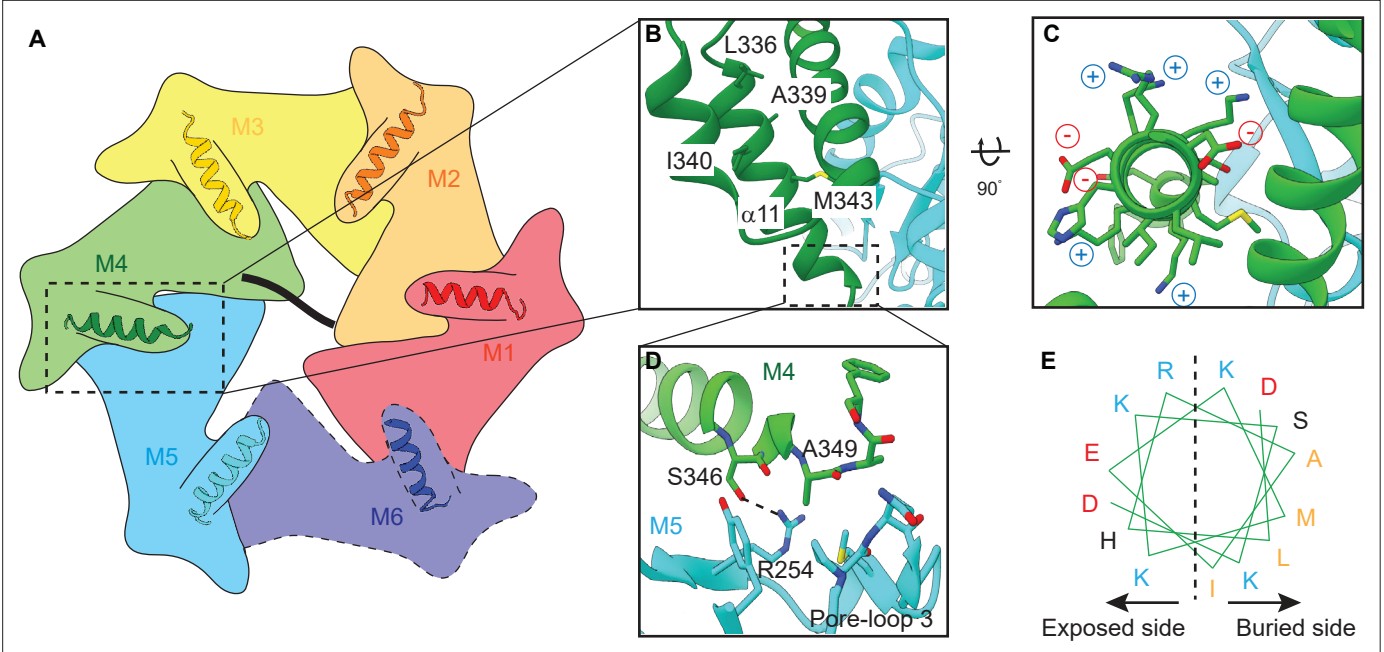

**Figure 2.** Helix α11 resides at the intersubunit interface of ATAD1. (**A**) Schematic showing the position of helix α11 within each ATAD1 subunit. ATAD1 subunits are colored as in *Figure 1*. α11 is shown in cartoon representation. Different from *Figure 1*, here, the cytosol-facing side (instead of the membrane-facing side) of ATAD1 is shown, thus the order of subunits from M1, M2 to M6 goes counterclockwise. (**B**) to (**D**) zoomed-in view of α11 interaction with the neighboring subunits. (**E**) Helical wheel showing the amphipathic property of α11. Hydrophobic amino acids are colored in orange, positive charged amino acids in blue, negative charged amino acids in red, and polar amino acids in purple.

The online version of this article includes the following figure supplement(s) for figure 2:

**Figure supplement 1.** Helices α11 form an additional spiral staircase beneath pore-loop 3.

**Figure supplement 2.** ATAD1 possesses a unique helix at the C-terminus that is structured differently from those in other AAA$_{MC}$ proteins.

asked whether this α-helix also mediates intersubunit contacts within the other subunits in the spiral. Indeed, slightly extended EM density for this α-helix exists in other subunits as well (*Figures 1C and 2A*). We modeled an α-helix (α11, amino acids 333–346) followed by a short turn (amino acids 347–351) into this density. Density beyond that (amino acids 352–361) was not clearly visible indicating its lack of rigid structure. A similar α-helix existed in the crystal structure of the monomeric *S.c.* Msp1 (*Wohlever et al., 2017*). However, the last few amino acids in that α-helix in the *S.c.* Msp1 were replaced by amino acids left from protease cleavage, making it difficult to interpret the original structure of this region.

In ATAD1's helical assembly, six α11s form a staircase beneath pore-loop 3, which does not contact the substrate directly but forms an interconnected network with pore-loops 1 and 2 that directly engage the substrate (*Figure 2—figure supplement 1*). Zooming in on α11 reveals that it lies at the interface between two adjacent subunits (*Figure 2A*). It forms extensive contacts with both the rest of the subunit and the *counterclockwise* (when viewed from the cytosol-facing side, as shown in Figure 4) adjacent subunit, bridging between the two (*Figure 2*). In α11$_{M4}$, for example, L336, A339, and I340 pack against the core of M4 (*Figure 2A*); M343 rests on the interface created by M4 and M5, and A349 points to a hydrophobic groove formed by pore-loop 3 of the M5. Finally, both the side chain and the backbone carbonyl of S346 form hydrogen bonds with R254 of M5 (*Figure 2D*). A few charged amino acids (R254, K342, K344, K345, K347, E341, and D348) point into the cytosol (*Figure 2C and E*).

Interestingly, other AAA$_{MC}$ proteins such as katanin, spastin, and Vps4 are structured differently in this region (*Han et al., 2020*; *Han et al., 2017*; *Sandate et al., 2019*; *Shin et al., 2019*). For those proteins, α11 is shorter (consisting of three helical turns instead of four as seen in ATAD1). It is followed by a loop and another short helix, α12, that reaches across the intersubunit interface to contact the *clockwise* adjacent subunit (*Figure 2—figure supplement 2*). α12 is functionally important as its deletion impacts oligomer assembly and protein function both in vitro and in vivo (*Vajjhala et al.,*

*2008*). Since ATAD1 lacks α12, and instead has a longer α11, we hypothesized that α11 could also mediate hexamer assembly. If so, we expected that a mutant version might fail to remove mislocalized membrane proteins in cells lacking wild-type (WT) ATAD1, but be unable to poison WT ATAD1's activity, as it would not be able to incorporate into WT hexamers. In other words, we expected it to display a recessive phenotype, rather than a dominant one. To test this notion, we next sought to establish an assay that distinguishes between dominant and recessive ATAD1 mutations and allows us to measure the activity of ATAD1 variants in a direct and quantifiable way.

## Direct visualization and quantification of ATAD1's activity in cells

One of ATAD1's established substrates is Gos28, a Golgi-localized TA SNARE protein. In cells lacking functional ATAD1, Gos28 localizes to mitochondria (*Chen et al., 2014*). We thus sought to use the localization of Gos28 as a readout for ATAD1's activity (*Figure 3A*). To analyze the activity of ATAD1 mutants in the absence of the WT enzyme, we deleted both alleles of *ATAD1* in HeLa cells using CRISPR/Cas9 (*Figure 3— figure supplement 1*). We next stably expressed Gos28 with an N-terminal EGFP tag in the ATAD1$^{-/-}$ cells and used confocal microscopy to visualize the localization of EGFP-Gos28. In ATAD1$^{-/-}$ cells, we observed two populations of Gos28 molecules: one visualized as an extended perinuclear region characteristic of the Golgi apparatus and the other in a spread-out network (*Figure 3B*, top row). Overlap of the EGFP signal with the signal from the mitochondria stain, MitoTracker Deep Red, showed that the latter population corresponds to the mitochondrial network, indicating that in ATAD1$^{-/-}$ cells, a portion of EGFP-Gos28 molecules are mislocalized to mitochondria, in line with a previous study (*Chen et al., 2014*). We then expressed WT ATAD1 labeled with a C-terminal HaloTag in the ATAD1$^{-/-}$ cells (*Figure 3B*, middle row) and observed a prominent shift of EGFP-Gos28 signal toward a Golgi-like distribution, indicating that ATAD1 cleared mislocalized EGFP-Gos28 from the mitochondria. Finally, we expressed the ATAD1 bearing a mutation that inactivates its ATPase activity, ATAD1$^{E193Q}$. The result mimicked the localization pattern seen in the ATAD1$^{-/-}$ cells (*Figure 3B*, top row), indicating that the ability of clearing mislocalized EGFP-Gos28 was dependent on ATAD1's enzymatic activity. We similarly observed ATAD1-dependent removal of mistargeted Pex26, a peroxisomal TA protein (*Matsumoto et al., 2003*), from mitochondria, suggesting that monitoring the clearance of a mislocalized protein is a reliable method to examine ATAD1-dependent TA protein extraction (*Figure 3—figure supplement 2*).

In addition to expressing the reporter in the ATAD1$^{-/-}$ cell line, we stably expressed Gos28 in a WT HeLa cell line. We expected ATAD1 variants that impact its ability to form proper oligomers to induce Gos28 mislocalization in the ATAD1$^{-/-}$ cell line, but not in the WT cell line. By contrast, inactive ATAD1 variants that retain their ability to assemble into hexamers should act as dominant inhibitors of the enzyme and induce substrate mislocalization.

To test the reliability of the system, we first expressed a known dominant-negative mutant, ATAD1$^{E193Q}$, within the WT reporter cells (*Figure 3—figure supplement 3*). Expression of ATAD1$^{E193Q}$ induced mislocalization of EGFP-Gos28 to mitochondria, indicating that ATAD1$^{E193Q}$ was incorporated into hexamers, poisoning WT activity. As expected, expression of the WT ATAD1 or an empty vector did not induce EGFP-Gos28 mislocalization in this background.

Before proceeding to test the function of α11, we wanted to see if we could quantify the degree of EGFP-Gos28 mislocalization in an unbiased fashion and use it as a readout for ATAD1's activity. To this end, we developed a data analysis pipeline using CellProfiler (*McQuin et al., 2018*; *Figure 3C*, *Figure 3—source data 1*). In brief, we first identified cells expressing ATAD1-HaloTag, using the cell-permeable JF549 dye (*Grimm et al., 2017*) to label and visualize Halo-tagged ATAD1. We then measured the colocalization of EGFP-Gos28 and mitochondria (stained with MitoTracker) in ATAD1-expressing cells using the Pearson correlation coefficient (PCC) as the metric, measured between the two channels for each cell (*Figure 3D*). As shown in *Figure 3D*, expression of an empty vector or ATAD1$^{E193Q}$ in the ATAD1$^{-/-}$ reporter cell line led to a PCC of 0.31±0.022 and 0.31±0.017, respectively. By contrast, transiently expressing WT ATAD1 showed a significantly lower PCC of 0.13±0.015 (p<0.0001), confirming the visually evident changes in EFGP-Gos28 localization and validating this approach for the evaluation of functional mutants of ATAD1.

With the EGFP-Gos28 reporter assay established in both an ATAD1$^{-/-}$ background and a WT background, we next used it to test α11's function. To this end, we made two truncated versions of ATAD1: one in which we deleted 30 amino acids at the C-terminus (removing α11 and everything C-terminal to it, ATAD1Δα11, *Figure 1C*), and another in which we kept α11 and removed everything C-terminal to it (ATAD1$_{1-351}$, *Figure 1C*). We first expressed these variants in the ATAD1$^{-/-}$ cells. As shown in *Figure 4A*,

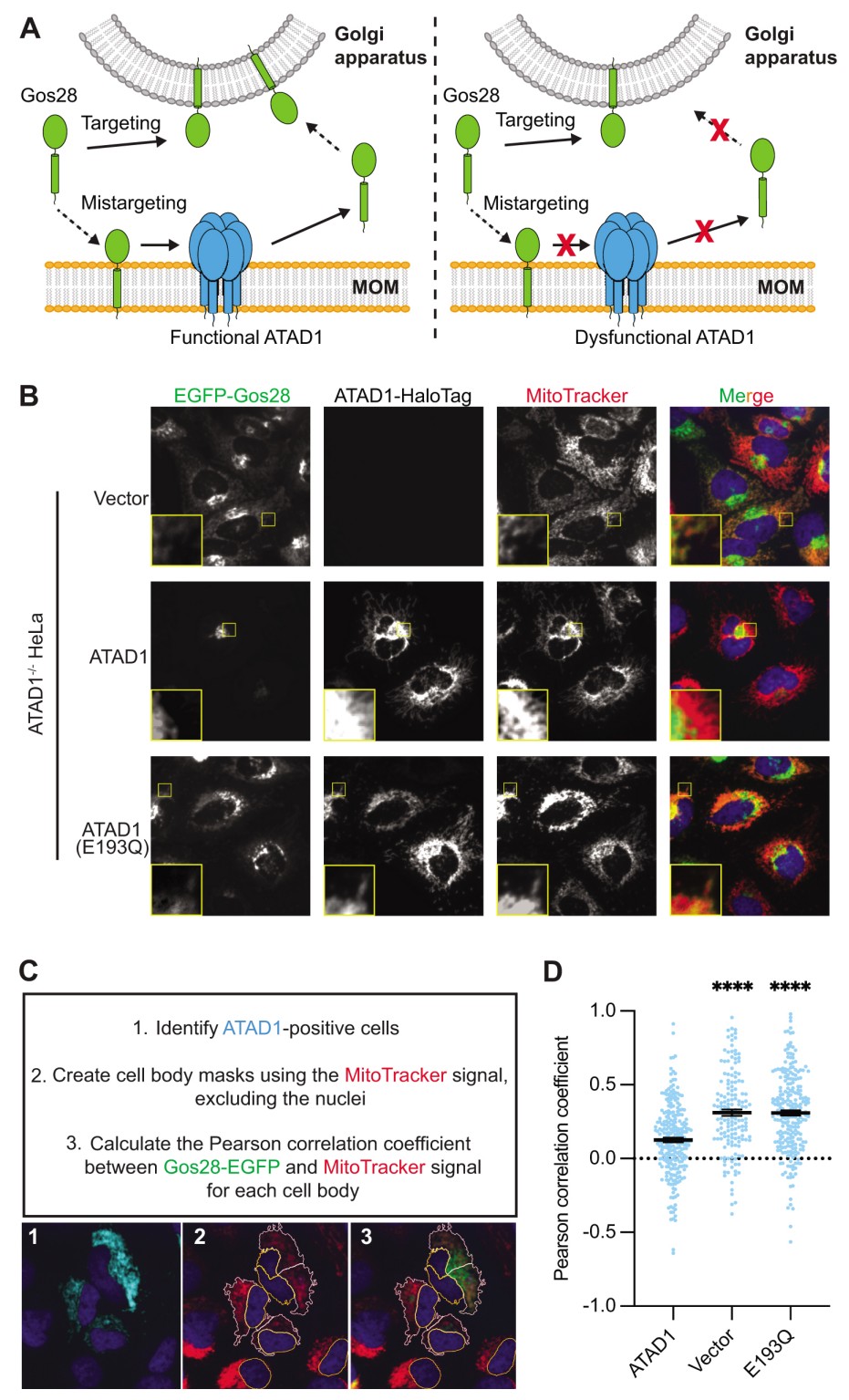

**Figure 3.** Microscopy assay allows for the direct measurement and quantification of ATAD1's activity. (**A**) Model for ATAD1-depedent extraction of Gos28: in cells expressing a functional ATAD1 (left panel), mislocalized Gos28 on the mitochondrial outer membrane (MOM) is extracted by ATAD1 and given a second chance to return to the Golgi apparatus. In cells expressing a dysfunctional ATAD1 (right panel), Gos28 accumulates on the MOM. (**B**) Representative average intensity projection images of live HeLa ATAD1$^{-/-}$ cells stably expressing EGFP-Gos28 (green channel) and transiently expressing empty vector (*top row*), ATAD1-HaloTag (*middle row*), and

*Figure 3 continued on next page*

*Figure 3 continued*

ATAD1(E193Q)-HaloTag (*bottom row*). Mitochondria are stained with MitoTracker (red channel). The individual channels are shown in black and white, and overlay of the EGFP and the MitoTracker channels are shown in the right-most column with Hoechst-stained nuclei in blue. Insets are included to better show the absence or presence of colocalization between EGFP-Gos28 and the mitochondria. (**C**) Workflow of the CellProfiler pipeline for measuring EGFP-Gos28 mislocalization. (**D**) Mean Pearson correlation coefficient (PCC) values and the SEM between EGFP-Gos28 and the mitochondria when expressing the indicated construct. Individual cell PCC values are represented as a single dot. Significance values were calculated using the Mann-Whitney test. ****p<0.0001.

*Figure 3—source data 1* describes the data analysis pipeline and the raw image files used in this pipeline are available on Dryad.

The online version of this article includes the following source data and figure supplement(s) for figure 3:

**Source data 1.** Data analysis pipeline for live-cell imaging results.

**Figure supplement 1.** Verification of ATAD1 knockout by Western blot.

**Figure supplement 2.** Live-cell imaging showing the ATAD1-dependent localization of EGFP-Pex26.

**Figure supplement 3.** Live-cell imaging showing the ATAD1-dependent localization of EGFP-Gos28 in wild-type (WT) HeLa cells.

EFGP-Gos28 showed significant mislocalization to mitochondria in cells expressing ATAD1Δα11 but not ATAD1$_{1-351}$, suggesting that the α11 helix is required for its function. By contrast, when expressed in the WT cells, ATAD1Δα11 did not induce significant mislocalization of EGFP-Gos28 (*Figure 4B*), indicating its recessive phenotype, that is, its lack of incorporation into and inactivating the WT hexamers present in these cells (the average values for the biological replicates are shown in *Figure 4—figure supplement 1A* and B). While expression of ATAD1Δα11 did not induce substrate mislocalization on average, it was evident by eye that a small population of WT cells expressing ATAD1Δα11 showed EFGP-Gos28 mislocalization (*Figure 4—figure supplement 2A*). We assume that this effect resulted from high ATAD1Δα11 expression in these outlier cells that compensated for ATAD1Δα11's reduced oligomerization ability and allowed assembly into WT hexamers. Indeed, we observed a positive correlation between ATAD1Δα11 expression level and the degree of Gos28 mislocalization: cells that expressed more ATAD1Δα11 showed a higher degree of Gos28 mislocalization (*Figure 4—figure supplement 2B*), suggesting that ATAD1Δα11 incorporated into WT hexamers and inactivated them, or that the ATAD1Δα11 subunits prevented the WT enzyme from forming stable hexamers. These data combined showed that α11 plays an important role in both subunit oligomerization and hexamer function.

To further establish α11's impact on ATAD1 oligomerization, we examined the elution profile of recombinantly expressed ATAD1 with and without the C-terminal helix. To this end, we purified the cytosolic domain of ATAD1 lacking the α11 helix (Δ40-ATAD1Δα11). On size exclusion, Δ40-ATAD1Δα11 eluted exclusively as a monomer. At similar concentrations, Δ40-ATAD1 formed oligomers (*Figure 4—figure supplement 3*), indicating that α11 mediates oligomerization in vitro. To evaluate the functional impact of a defect in hexamer formation, we next asked if the ability to bind a peptide substrate was affected for Δ40-ATAD1Δα11. To this end, we sought to identify a peptide that binds ATAD1. First, we designed a peptide array by stepping through ATAD1's known substrates and measured the binding of individual peptides to ATAD1. We picked a few peptides that showed the strongest binding and tested their affinity to ATAD1 in solution using fluorescence polarization. Out of this group, a peptide (P13) that is derived from the sequence of the C-terminus of Pex26 (an established ATAD1 substrate) emerged as the strongest binder, which we then used to test the effect of truncating α11 on ATAD1's substrate-binding affinity. As expected and consistent with its inability to form hexamers, Δ40-ATAD1Δα11 also showed substantially reduced peptide-binding affinity (*Figure 4C*).

Several disease-related mutations are close to the α11 region at the C-terminus (see Introduction). We tested these ATAD1 variants using our imaging assay to assess if their ability to remove mislocalized TA proteins would be impacted. Out of the variants tested, the E267stop variant impacted ATAD1's function significantly, whereas the other variants (H357Rfs*15, D221H, R9H, and E290K) displayed WT-like phenotypes (*Figure 4—figure supplement 4*). These results indicate that the mechanism that underlies neurological disorders may be separate from the extraction of Gos28.

As previously mentioned, α11 packs against pore-loop 3 from the adjacent subunit and constitutes the additional layer of the interconnected network involving three pore-loops (*Figure 2—figure supplement 1*). The fact that it lies at the oligomerization interface and also contacts pore-loop 3 prompted us to

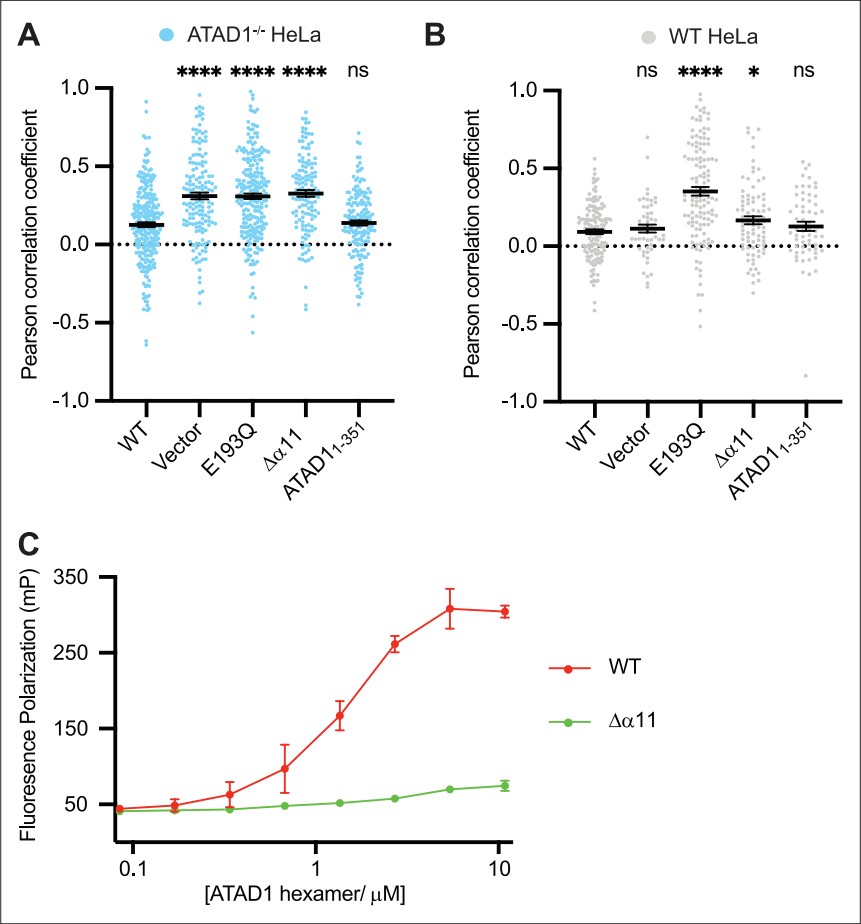

**Figure 4.** Helix α11 mediates hexamer assembly. (**A**) Mean Pearson correlation coefficient (PCC) values and the SEM between EGFP-Gos28 and the mitochondria in ATAD1$^{-/-}$ HeLa cells, transiently expressing controls or the ATAD1Δα11 mutant. (**B**) Mean PCC values and the SEM between EGFP-Gos28 and the mitochondria in wild-type (WT) HeLa cells, transiently expressing controls or the ATAD1Δα11 mutant. Significance values were calculated using the Mann-Whitney test. *p<0.05, ****p<0.0001. The raw image files used for this analysis are available on Dryad. (**C**) Fluorescence polarization assay showing the different peptide-binding abilities of the Δ40-ATAD1 (WT) versus Δ40-ATAD1Δα11 (Δα11). ATAD1 concentrations are expressed as hexamer concentrations. Each dot represents an average of two replicates and the error bar represents SEM.

The online version of this article includes the following figure supplement(s) for figure 4:

**Figure supplement 1.** Reproducibility of EGFP-Gos28 mislocalization in live-cell imaging with expression of ATAD1Δα11.

**Figure supplement 2.** ATAD1Δα11 expression level impacts substrate mislocalization in wild-type (WT) HeLa cells.

**Figure supplement 3.** Size exclusion chromatography (SEC) of Δ40-ATAD1 and Δ40-ATAD1Δα11.

**Figure supplement 4.** Testing the effect of the disease-relevant mutations of ATAD1.

**Figure supplement 5.** ATPase assay of Δ40-ATAD1 and Δ40-ATAD1Δα11.

ask whether α11 could couple oligomerization with substrate binding. Given that oligomerization usually enhances the ATPase activity of AAA proteins, we sought to measure the ATPase activity of Δ40-ATAD1 and Δ40-ATAD1Δα11 and asked whether the activities can be stimulated by the addition of substrate. As shown in *Figure 4—figure supplement 5*, Δ40-ATAD1Δα11 displays a much lower ATPase activity compared to Δ40-ATAD1, indicating the lack of functional hexamers. With the addition of a peptide substrate (P13), the activity of Δ40-ATAD1 is stimulated by around 41%, whereas no significant stimulation was observed for Δ40-ATAD1Δα11 (*Figure 4—figure supplement 5*). This results suggest that Δ40-ATAD1Δα11's ability to oligomerize is significantly impacted and the addition of a substrate is not able to bring the monomeric subunits into a functional oligomeric form. Having examined the oligomer interface in detail, we next examined the substrate interactions within the central pore.

## The highly aromatic central pore of ATAD1 is crucial for substrate binding and extraction

Upon building all of ATAD1's amino acids into the EM density, we observed a piece of density in the central pore that resembles a linear peptide in both the closed and the open state structures. This density likely represents a composite of peptides that co-purified with ATAD1 from *Escherichia coli*. The side chain density was clearly visible, and we modeled the peptide as a 10-mer polyalanine. A peptide in the corresponding position was also observed in the Msp1 structures (**Wang et al., 2020**), suggesting the proteins' high affinity for their substrates.

Like for Msp1, six ATAD1 subunits tightly surround the peptide in the central pore. As summarized in the introduction, amino acids used by ATAD1 to contact the substrate are phylogenetically conserved within the Msp1/ATAD1 family, but differ from other AAA proteins. Each ATAD1 subunit extends two short loops containing three aromatic amino acids to engage the substrate peptide (**Figure 1D and E**). Six pore-loops 1 form a spiral staircase surrounding the peptide substrate. Pore-loop 1 is comprised of a conserved $KX_1X_2G$ motif, where $X_1$ is a tryptophan (W166) and $X_2$ a tyrosine (Y167). Both aromatic amino acids intercalate into the side chains of the translocating peptide (**Figure 1D**), holding it in place. Below pore-loop 1 is a second staircase formed by pore-loops 2, which use a histidine (H206) to contact the backbone of the substrate (**Figure 1E**). By contrast, in most other known AAA proteins, the $X_1$ position within pore-loop 1 is an aromatic amino acid, and the $X_2$ position is an aliphatic amino acid that does not contact the substrate. Also, their pore-loops 2 usually do not contact the substrate directly.

## Multiple aromatic amino acids in the central pore are essential for ATAD1's function

We next revisited our hypothesis that ATAD1 function, like Msp1 function, necessitates engagement of multiple aromatic amino acids in its central pore. To this end, we tested the effect of pore-loop mutations in the ATAD1$^{-/-}$ EGFP-Gos28 reporter cell line (**Figure 5A**) using our cell-based assay. Expression of ATAD1 bearing either the W166A (the first aromatic amino acid in pore-loop 1) or Y167A (the second aromatic amino acid in pore-loop 1) mutation led to significant mislocalization of EGFP-Gos28, suggesting that these mutations inactivate ATAD1. By contrast, expression of the aromatic mutant Y167F cleared Gos28 from mitochondria, indicating functional ATAD1, which is consistent with what we observed for Msp1. Interestingly, changing W166 or Y167 to an aliphatic amino impacted ATAD1's activity: ATAD1$^{W166V}$, ATAD1$^{Y167V}$, and ATAD1 $^{Y167L}$ were inactive. The activity of ATAD1$^{W166L}$ was also impacted, albeit to a lesser degree (**Figure 5A**, **Figure 5—figure supplements 1 and 2**), supporting the notion that the aromaticity of this position is important. This observation, namely that an aliphatic amino acid did not functionally replace the aromatic amino acid in this position, contrasts with our data from corresponding mutations in Msp1 in the yeast growth assay (**Wang et al., 2020**), suggesting either a better sensitivity of the microscopy-based assay or an inherit difference between the yeast and the mammalian system. We also tested the effect of mutations on H206 (pore-loop 2) in our assay. While our structural data clearly showed an interaction between H206 and the substrate backbone, mutations in this position did not impact the function of ATAD1 in this context.

Having established the functional importance of the pore-loop 1 side chains in substrate extraction, we further examined their specific impact on ATAD1's ability to bind a substrate in vitro. Consistent with the cell-based assay, ATAD1$^{W166A}$, ATAD1$^{W166V}$, ATAD1$^{W166L}$, ATAD1$^{Y166A}$, and ATAD1$^{Y167V}$ showed dramatically reduced (estimated >100-fold) binding to P13 compared to WT ATAD1. Both ATAD1$^{H206A}$ and ATAD1$^{Y167F}$ showed WT-like activities in the cell-based assay, but bound P13 with much weaker affinity than the WT enzyme in this assay. We reasoned that this difference arose either because the in vitro assay is more sensitive to small changes in activity that cell-based assays fail to capture, or because a different substrate was used in the cell-based assay (Gos28) than in the in vitro binding assay (P13, derived from Pex26). Taken together, the in vitro peptide-binding results, implications from the structure, and the cell-based assay all converge on the conclusion that the aromaticity of the second amino acid in pore-loop 1 is important for ATAD1's function.

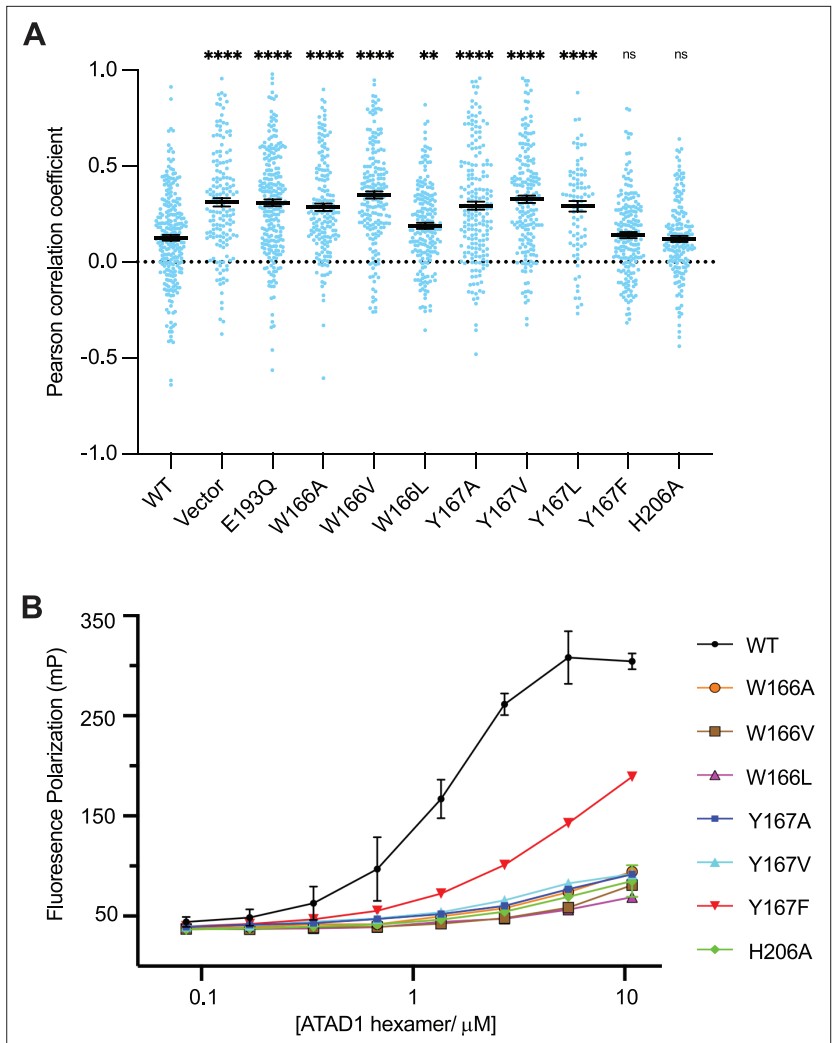

**Figure 5.** Pore-loop 1 aromatic amino acids are important for ATAD1's function both in vivo and in vitro. (**A**) Mean Pearson correlation coefficient (PCC) values and the SEM between EGFP-Gos28 and the mitochondria in cells expressing ATAD1 variants. Individual cell PCC values are represented as a single dot. Significance values were calculated relative to the wild-type (WT) condition using the Mann-Whitney test. **p<0.01, ****p<0.0001. The raw image files used for this analysis are available on Dryad. (**B**) Different ATAD1 variants' peptide-binding abilities as measured by fluorescence polarization. ATAD1 concentrations are expressed as hexamer concentrations. Here, wild-type (WT) refers to Δ40-ATAD1, and each mutant refers to Δ40-ATAD1 bearing that mutation. Each dot represents the average of two replicates, and the error bar represents the SEM.

The online version of this article includes the following figure supplement(s) for figure 5:

**Figure supplement 1.** Live-cell imaging showing the pore-loop dependent localization of EGFP-Gos28 in ATAD1$^{-/-}$ HeLa cells.

**Figure supplement 2.** Reproducibility of EGFP-Gos28 mislocalization in live-cell imaging with expression of pore-loop mutants.

## Discussion

In this work, we present the cryo-EM structures of the soluble domain of the mitochondrial AAA protein ATAD1 in complex with a peptide substrate. Our structures show that the overall architecture of ATAD1 is very similar to that of its yeast homolog Msp1. ATAD1's architecture fully agrees with the conserved mechanism of hand-over-hand spiral propagation established for many AAA proteins in substrate translocation. We also discovered a distinct structural element, helix α11, that was present but remained structurally unresolved in Msp1, which differentiates ATAD1/Msp1 from other AAA$_{MC}$

proteins. Multiple lines of evidence including cell-based mislocalization, substrate binding, and ATPase activity all suggest that α11 is required for the formation of functional oligomers. Although we do not know why it is structured differently than the other AAA$_{MC}$ proteins (which adopt an α11–α12 organization), one possibility is that α11 is useful in adaptor interaction. In particular, yeast Msp1 docks to the proteosome to channel its substrate directly for degradation (*Basch et al., 2020*). Another study showed that during mitochondrial import stress, Msp1 is recruited to the TOM complex by the adaptor Cis1 to remove precursor proteins stuck in the TOM translocase (*Weidberg and Amon, 2018*). We reason that structural elements facing the cytosolic side of Msp1/ATAD1 (such as α11) may be evolutionarily adapted to interact with these factors. In this view, the α11–α12 organizational differences in different AAA$_{MC}$ protein subfamilies may have evolved to suit their specialized functions, such as binding to microtubules and ESCRT proteins for katanin and Vps4, respectively. This view is supported by the fact that ATAD1/Msp1 occupy the basal position in the evolutionary tree of this clade, and the other members (katanin, spastin, and Vps4) share a common evolutionary ancestor (*Frickey and Lupas, 2004*).

The cell-based imaging assay allowed us to directly quantify ATAD1-dependent substrate mislocalization and thereby revisit a hypothesis raised in the previous study (*Wang et al., 2020*) on the functional importance of the pore-loop amino acids. The notion that aromaticity of pore-loop/substrate-interacting side chains is important is supported by single molecule force measurements of other AAA protein translocases. A study on the bacterial AAA proteins ClpXP (*Rodriguez-Aliaga et al., 2016*), for example, showed that the flat, bulkier pore-loop amino acids give the protein a stronger grip, that is higher coupling efficiency between ATP hydrolysis and substrate unfolding, yet are compromised in the rate of substrate translocation (measured as protein unfolding with ClpXP). Given ATAD1's function is to extract hydrophobic membrane proteins from the lipid membrane, having a tighter grip on the substrate may be important, even if it comes at the cost of a slower translocation speed. Similar to ATAD1, Cdc48, a protein that removes misfolded membrane proteins from the ER membrane, also possesses two aromatic amino acids in its pore-loop 1 (*Cooney et al., 2019*; *Twomey et al., 2019*), suggesting that the additional aromatic amino acids in the central pore evolved to aid in the removal of membrane proteins.

To conclude, while ATAD1/Msp1 utilize a conceptually similar mechanism for substrate translocation as many other AAA proteins, the high conservation of several unique features between ATAD1 and Msp1 suggests that evolution fine-tuned these enzymes early in eukaryotic cell evolution for their special role in membrane protein extraction and protein quality control. The phylogenetic comparison of structure and function, and the complementary experimental opportunities afforded by structural and cell-level analysis, allowed us to extract hints regarding important functional principles and assess their generality over a wide span of evolutionary time. The neurological phenotypes associated with a number of ATAD1 mutations still remain poorly understood, yet they serve to underscore the importance of proteostasis in human physiology and raise hope that understanding these principles may help to develop new treatments for combating devastating pathological dysfunctions arising from proteostasis imbalances.

# Materials and methods

**Key resources table**

| Reagent type (species) or resource | Designation | Source or reference | Identifiers | Additional information |
|---|---|---|---|---|
| Gene (*Homo sapiens*) | ATAD1 | IDT | | For human expression |
| Gene (*Homo sapiens*) | ATAD1 | IDT | | Codon-optimized for *E. coli* expression |
| Gene (*Homo sapiens*) | EGFP-GOS28 | TwistBioscience | | |
| Gene (*Homo sapiens*) | EGFP-PEX26 | TwistBioscience | | |

*Continued on next page*

*Continued*

| Reagent type (species) or resource | Designation | Source or reference | Identifiers | Additional information |
|---|---|---|---|---|
| Strain, strain background (*Escherichia coli*) | BL21(DE3) | Life Technologies | Cat# C601003 | |
| Cell line (*Homo sapiens*) | HeLa | ATCC | | |
| Cell line (*Homo sapiens*) | HeLa, ATAD1KO | Synthego | | |
| Antibody | Anti-ATAD1 (Mouse, monoclonal) | NeuroMab | Cat# N125/10 | WB (1:500) |
| Antibody | Anti-Histone H3 (Rabbit, polyclonal) | Abcam | Cat# ab1791 | WB (1:1500) |
| Recombinant DNA reagent | 6xHis-thrombin-ATAD1$^{E193Q}$ (plasmid) | This paper | | Materials and methods section 'Molecular cloning' |
| Recombinant DNA reagent | CMVd3-ATAD1-HaloTag | This paper | | Materials and methods section 'Molecular cloning' |
| Recombinant DNA reagent | CMV-GOS28-EGFP | This paper | | Materials and methods section 'Molecular cloning' |
| Recombinant DNA reagent | CMV-PEX26-EGFP | This paper | | Materials and methods section 'Molecular cloning' |
| Peptide, recombinant protein | P13 | GenScript | | 5-FAM-FSRLYQLRIR |
| Commercial assay or kit | In-Fusion Snap Assembly Master Mix | Takaro Bio | Cat# 638948 | |
| Commercial assay or kit | QuikChange II XL Site-Directed Mutagenesis Kit | Agilent | Cat# 200521 | |
| Chemical compound, drug | Janelia Fluor 549 dye | Janelia Fluor Dyes | | Gift of Dr Luke Lavis |
| Chemical compound, drug | MitoTracker Deep Red FM | Life Technologies | Cat# M22426 | |
| Chemical compound, drug | Hoechst 33342 | Life Technologies | Cat# H1399 | |
| Software, algorithm | CellProfiler 4.0.5 | https://doi.org/10.1371/journal.pbio.2005970 | | |
| Software, algorithm | MotionCor2 | PMID:28250466 | RRID: SCR_016499 | |
| Software, algorithm | Relion | PMID:23000701 | RRID: SCR_016274 | |
| Software, algorithm | CryoSPARC | PMID:28165473 | RRID: SCR_016501 | |
| Software, algorithm | UCSF Chimera | PMID:15264254 | RRID: SCR_004097 | |
| Software, algorithm | GCTF | PMID:26592709 | RRID: SCR_016500 | |
| Software, algorithm | Phenix | PMID:20124702 | RRID: SCR_014224 | |
| Software, algorithm | Coot | PMID:20383002 | RRID: SCR_014222 | |

## Molecular cloning

To generate the construct used for cryo-EM studies, the gene encoding the cytosolic domain of human Δ40-ATAD1 was PCR amplified and subcloned into a pET28 vector encoding an N-terminal 6xHis tag followed by a thrombin cleavage site. The Walker B mutation (E214Q) was introduced by Quick-Change site-directed mutagenesis. To generate the constructs used for imaging, the gene blocks for the human *ATAD1* was synthesized by IDT; the gene blocks for *Gos28* and *Pex26* were synthesized by Twist Bioscience. *Gos28* or *Pex26* was C-terminally fused to *EGFP* and cloned downstream of a CMV promoter within a lentivirus production vector. *ATAD1* was N-terminally fused to *HaloTag* and cloned downstream of a truncated CMV promoter (CMVd3) for transient transfections. Mutations to *ATAD1* were made by QuickChange site-directed mutagenesis. All the constructs are verified by Sanger sequencing.

## Protein purification

His-Δ40-ATAD1$^{E193Q}$ was expressed and purified as previously described for His-Δ30-Msp1$^{E214Q}$ (*Wang et al., 2020*).

## Sample preparation of cryo-EM

His-Δ40-ATAD1[E193Q] was diluted to around 100 μM in buffer containing 25 mM HEPES pH 7.5, 300 mM NaCl, 1 mM DTT, 2.5% glycerol, 2 mM ATP, and 2 mM $MgCl_2$. The sample was incubated on ice for 1–2 hr before plunge freezing. A 3 μl aliquot of the sample were applied onto the Quantifoil R 1.2/1/3 400 mesh Gold grid and incubated for 15 s. A 0.5 μl aliquot of 0.1–0.2% Nonidet P-40 substitutes was added immediately before blotting. The entire blotting procedure was performed using Vitrobot Mark IV (FEI) at 10°C and 100% humidity.

## EM data collection

Cryo-EM data was collected on a Titan Krios transmission electron microscope operating at 300 keV and micrographs were acquired using a Gatan K3 summit direct electron detector. The total dose was 60 e⁻/Å² ($60\ e^-/Å^2$), fractioned over 100 frames during a 10 s exposure. Data was collected at 105,000× nominal magnification (0.832 Å/pixel at the specimen level) and nominal defocus range of –1.0 to –2.0 μm.

## Cryo-EM data analysis

Micrograph frames were aligned using MotionCorr2. The contrast transfer function (CTF) parameters were estimated with GCTF (*Zhang, 2016*). Particles were automatically picked using Gautomatch and extracted in RELION (*Scheres, 2012*) using a 320-pixel box size. Images were down-sampled to a pixel size of 3.328 Å and classified in 2D in RELION. Classes that showed clear protein features were selected and extracted with re-centering and then subjected to 3D classification. Particles from the best class emerging from 3D classification were then subjected to 2D classification followed by another round of 3D classification to further purify the particles. Particles that showed clear hexameric features were then re-extracted (pixel size = 0.832 Å/pixel) and imported into cryoSPARC (*Punjani et al., 2017*). Within cryoSPARC, particles were subjected to another round of 2D classification followed by heterogenous refinement, from which two distinct conformations (the open versus the closed conformations) were discovered. CTF refinement followed by nonuniform refinement was performed on both conformations to yield final reconstructions of at 3.2 and 3.5 Å, for the closed and the open conformations. The final reconstructed maps are deposited into the EMDB under the accession codes EMD-26674 (closed conformation) and EMD-26675 (open conformation).

## Atomic model building and refinement

Model building and refinement was done in a similar way as previously described (*Wang et al., 2020*). Briefly, the big and the small AAA domain of the crystal structure of the monomeric *S.c.* Msp1 (*Wohlever et al., 2017*) was used to generate the predicted structures of the human ATAD1 in SWISS-MODEL (*Waterhouse et al., 2018*). The six big AAA domains and the six small AAA domains were individually docked into the map of His-Δ40-ATAD1[E193Q] in *UCSF Chimera* (*Pettersen et al., 2004*) using the *Fit in Map* function. The resulting model was subjected to rigid body refinement in Phenix (*Adams et al., 2010*), followed by real space refinement in *Coot* (*Emsley et al., 2010*). After the protein part has been modeled, a piece of continuous density was left in the central pore, into which we modeled a polyalanine sequence. Significant density was visible in the nucleotide-binding pockets within subunits M1 through M5, and an ATP molecule was modeled into that density. For the M6 subunit, in the open conformation the density there was not clear enough to distinguish between ATP and ADP, so although an ADP molecule was modeled, we indicated in *Figure 1* that it could be either. In the closed conformation, no significant density was observed in the nucleotide-binding pocket. The figures displaying structures were prepared with *UCSF Chimera*. The final models are deposited into the protein data bank (PDB) under the accession codes 7UPR (closed conformation) and 7UPT (open conformation).

## Fluorescence polarization

All fluorescence polarization experiments were done in the FP assay buffer containing 25 mM HEPES (pH 7.5), 150 mM KCl, 2 mM $MgCl_2$, 1 mM DTT, and 2 mM ATPγS, and measured in 384-well non-stick black plates using ClarioStar PLUS (BMG LabTech) at room temperature. Prior to the reaction setup, ATAD1 was diluted in twofold dilution series and incubated with 100 nM fluorescently labeled peptide (P13: 5-FAM-FSRLYQLRIR, purchased from Genscript) for 20 min at room temperature. Then, the

mixture was subjected to measurement of parallel and perpendicular intensities (excitation: 482 nm, emission: 530 nm). Data was plotted using GraphPad Prism 8.

## ATPase assay

ATPase activity of ATAD1 was measured using the oxidation of NADH as a readout of ATP hydrolysis. An enzyme mixture containing 0.2 mM NADH, 1 mM phosphoenol-pyruvate (PEP), 50 U/ml of pyruvate kinase, and lactate dehydrogenase was added to the wells of a 384-well black plate (Corning). Ten µM ATAD1, assay buffer (25 mM HEPES pH 7.5, 100 mM KCl, 2 mM $MgCl_2$, 10 µM BSA, 0.05% Tween-20), and either 50 µM of the unlabeled version of P13 peptide (FSRLYQLRIR) or a blank was added to the enzyme mixture. Samples were incubated for 20 min at 37°C before 1 mM ATP was added to start the reactions. Absorbance at 340 nm was measured every 15 s for a total of 60 min using the CLARIOstar Plus (BMG LabTech) microplate reader. Data was plotted using GraphPad Prism 8.

## Cell culture and transduction

HeLa cells were cultured in DMEM supplemented with 10% FBS, 100 U/ml penicillin/streptomycin, and 6 mM L-glutamine. A pooled ATAD1$^{-/-}$ cell population was generated by Synthego with a guide RNA targeting Exon 5 (CGGUCAGUGUCGAAGGCUGA). Monoclonal populations were obtained using limiting dilution in a 96-well plate and expanding single cell populations into a six-well plates. Knockouts were confirmed by Sanger sequencing and Western blotting using anti-ATAD1 antibody (N125/10, NeuroMab) and anti-Histone H3 (ab1791, Abcam) as a loading control. WT HeLa cells are from ATCC. HeLa cells with ATAD1 knockout (pooled) comes from Synthego. Both cell lines test negative for mycoplasma.

WT HeLa and ATAD1$^{-/-}$ HeLa cell lines expressing the reporter EGFP-Gos28 were generated by lentiviral infection. In brief, vesicular stomatitis virus (VSV)-G pseudotyped lentiviral particles were produced in 293METR packaging cells (kind gift of Brian Ravinovich, formerly at MD Anderson Cancer Center, Camden, NJ) using standard protocols. WT and ATAD1$^{-/-}$ HeLa cells were infected with concentrated virus (supplemented with 8 µg/ml polybrene) by centrifugal inoculation at 2000 rpm for 2 hr. Viral supernatant was removed following overnight incubation and cells were expanded for FACS. EGFP-positive cells were sorted using SONY SH800 FACS into high and low EGFP-expressing populations. For all imaging experiments, the population that has high EGFP expression was used.

## EGFP-Gos28 imaging

Expression of the ATAD1 variants in the cell-based assay was done through transient transfections. The day before transfections, cells were seeded on Ibidi 8-well glass bottom µ-slides in FluoroBrite DMEM (Life Technologies) media supplemented with glutamine and 10% FBS. Plasmids with the ATAD1 variants under a truncated CMVd3 promoter were transfected into cells using the FuGENE HD transfection reagent (Promega), following the manufacturer's protocol. Cells were incubated for 48 hr before imaging. For visualizing cell structures, nuclei were stained with Hoechst 33342 and mitochondria were stained with MitoTracker Deep Red FM (ThermoFisher). Transfected cells were stained with 25 nM Janelia Fluor 549 dye conjugated with the HaloTag ligand (JF549-HaloTag; kind gift of Dr Luke Lavis). Cells were incubated with the dyes for 15 min at 37°C followed by three washes with FluoroBrite DMEM media.

## Microscopy

Confocal imaging was carried out on a Nikon Ti-E inverted microscope equipped with a Yokogawa CSU-X high-speed confocal scanner unit and an Andor iXon 512×512 EMCCD camera. All images were acquired through a 40×1.3 NA oil immersion objective. Images were typically acquired with 60×EM gain and 100 ms exposure. The four lasers used were 405 nm (operated at 10 mW), 488 nm (operated at 25 mW), 561 nm (operated at 25 mW), and 640 nm (operated at 15 mW). All components of the microscope were controlled by the µManager open-source platform (*Edelstein et al., 2010*). The microscope stage was enclosed in a custom-built incubator that maintained preset temperature and $CO_2$ levels for prolonged live-imaging experiments. To avoid unintentional selection bias, fields-of-view were selected by only looking at stained cell nuclei in the 405 nm channel. No cells or fields-of-view were subsequently excluded from analysis, ensuring that the data faithfully capture the distribution of fluorescence across the entire cell population.

## Image quantification with CellProfiler

For each experiment, 15 fields-of-view were imaged per condition with an average number around 50 cells total. Average intensity z-projections were made for each image in each of the four channels and used as the input for the CellProfiler pipeline (*McQuin et al., 2018*). For automated image analysis, we developed a pipeline in CellProfiler 4.0.5. First, the images were background-corrected in every channel. Then, nuclei were identified as starting points for the propagation of cell body masks. Cells in which integrated ATAD1 signal (labeled with the JF549-HaloTag dye) passed a manually chosen threshold were identified as ATAD-positive cells. Integrated ATAD1 signal intensity was used as a proxy for ATAD1 expression level in each given cell. Within those cells, the MitoTracker signal was used to create a pseudo-cell boundary mask, closing all gaps. This serves as a reasonably good proxy for total cell area because the mitochondrial network is broadly distributed throughout HeLa cells. Measurements for colocalization were made inside this mask and outside of the nucleus. PCCs were calculated for each individual cell by performing a pixel-wise comparison of the EGFP and MitoTracker channels. The data analysis pipeline is described in *Figure 3—source data 1*.

## Acknowledgements

We thank the Walter lab for helpful discussions throughout the course of this project; H Zhou and Z Yu of the Cryo-EM facility at the HHMI Janelia Research Campus. We thank the QB3 shared cluster for computational support. We also thank Hehua, and Obie for their constant support throughout this project. This work was supported by the National Institutes of Health (R01GM032384). Lan Wang is a Damon Runyon Cancer Research Foundation fellow supported by the Damon Runyon Cancer Research Foundation (DRG-2312-17), Vladislav Belyy is a Damon Runyon Fellow supported by the Damon Runyon Cancer Research Foundation (DRG-2284-17), and Peter Walter is an Investigator of the Howard Hughes Medical Institute.

## Additional information

### Funding

| Funder | Grant reference number | Author |
|---|---|---|
| National Institutes of Health | R01GM032384 | Lan Wang<br>Hannah Toutkoushian<br>Vladislav Belyy<br>Peter Walter |
| Damon Runyon Cancer Research Foundation | DRG-2312-17 | Lan Wang |
| Damon Runyon Cancer Research Foundation | DRG-2284-17 | Vladislav Belyy |

The funders had no role in study design, data collection and interpretation, or the decision to submit the work for publication.

### Author contributions

Lan Wang, Hannah Toutkoushian, Conceptualization, Data curation, Formal analysis, Investigation, Methodology, Visualization, Writing – original draft, Writing – review and editing; Vladislav Belyy, Conceptualization, Data curation, Formal analysis, Investigation, Methodology, Visualization, Writing – review and editing; Claire Y Kokontis, Data curation, Formal analysis, Methodology, Visualization, Writing – review and editing; Peter Walter, Conceptualization, Data curation, Formal analysis, Funding acquisition, Investigation, Methodology, Supervision, Validation, Visualization, Writing – original draft, Writing – review and editing

### Author ORCIDs

Lan Wang  http://orcid.org/0000-0002-8931-7201
Hannah Toutkoushian  http://orcid.org/0000-0002-7461-2005
Claire Y Kokontis  http://orcid.org/0000-0001-9397-651X

Peter Walter  http://orcid.org/0000-0002-6849-708X

**Decision letter and Author response**
Decision letter https://doi.org/10.7554/eLife.73941.sa1
Author response https://doi.org/10.7554/eLife.73941.sa2

## Additional files

### Supplementary files
• Transparent reporting form

### Data availability
All data generated or analyzed during this study are included in the manuscript and supporting files. The imaging analysis pipeline is described in Figure 3-source data 1. The raw images used for data analysis are deposited into dryad (https://doi.org/10.5061/dryad.q2bvq83n3). The final models are deposited into the protein data bank (PDB) under the accession codes 7UPR (closed conformation) and 7UPT (open conformation). The final reconstructed maps are deposited into the EMDB under the accession codes EMD-26674 (closed conformation) and EMD-26675 (open conformation).

The following datasets were generated:

| Author(s) | Year | Dataset title | Dataset URL | Database and Identifier |
|---|---|---|---|---|
| Wang L, Toutkoushian H, Belyy V, Kokontis CY, Walter P | 2022 | Conserved structural elements specialize ATAD1 as a membrane protein extraction machine | https://doi.org/10.5061/dryad.q2bvq83n3 | Dryad Digital Repository, 10.5061/dryad.q2bvq83n3 |
| Wang L, Toutkoushian H, Belyy V, Kokontis CY, Walter P | 2022 | Human mitochondrial AAA protein ATAD1 (with a catalytic dead mutation) in complex with a peptide substrate (closed conformation) | https://www.rcsb.org/structure/7UPR | RCSB Protein Data Bank, 7UPR |
| Wang L, Toutkoushian H, Belyy V, Kokontis CY, Walter P | 2022 | Human mitochondrial AAA protein ATAD1 (with a catalytic dead mutation) in complex with a peptide substrate (open conformation) | https://www.rcsb.org/structure/7UPT | RCSB Protein Data Bank, 7UPT |
| Wang L, Toutkoushian H, Belyy V, Kokontis CY, Walter P | 2022 | Human mitochondrial AAA protein ATAD1 (with a catalytic dead mutation) in complex with a peptide substrate (closed conformation) | https://www.ebi.ac.uk/emdb/EMD-26674 | Electron Microscopy Data Bank, EMD-26674 |
| Wang L, Toutkoushian H, Belyy V, Kokontis CY, Walter P | 2022 | Human mitochondrial AAA protein ATAD1 (with a catalytic dead mutation) in complex with a peptide substrate (open conformation) | https://www.ebi.ac.uk/emdb/EMD-26675 | Electron Microscopy Data Bank, EMD-26675 |

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
