## [Editor Report]

This study reports the cryo-EM structure of human ATAD1 (Msp1 in yeast), a AAA protein responsible for the extraction of mistargeted tail-anchored (TA) proteins from the mitochondrial outer membrane. The structure helps to understand the effects of disease-linked mutations on ATAD1/Msp1 activity.

---

## [Decision Letter]

**Decision letter after peer review:**

Thank you for submitting your article "Conserved structural elements specialize ATAD1 as a membrane protein extraction machine" for consideration by *eLife*. Your article has been reviewed by 3 peer reviewers, and the evaluation has been overseen by a Reviewing Editor and Volker Dötsch as the Senior Editor. The following individual involved in review of your submission has agreed to reveal their identity: Marta Carroni (Reviewer #3).

Essential revisions:

Main points:

1. Results of biological repeats should be presented for the microscopic mislocalization assay. It appears that the data shown is from a single experiment with multiple fields taken.

Also, if at all possible, an additional substrate protein should be tested. Considering the similarity in structural observations in this work and the last Msp1 paper, this additional cellular/reporter data would make the current work stand out.

2. Figure 2: The representation of subunits is different from Figure 1. This leads to substantial confusion with respect to the relative order of individual subunits. While in Figure 1 M1 is the clockwise subunit with respect to M6, it becomes the counterclockwise one in Figure 2. The depiction in Figure 2 seems unusual. M1 represents the top subunit in the spiral arrangement of ATAD1 subunits and the seam subunit M6 is on its way to become the new top subunit (new M1) during subunit cycling. Therefore, M1 is typically defined as clockwise subunit (see Figure 1) but not as the counterclockwise one (see Figure 2). The authors need to standardize the respective figures and data description and relate their findings to other AAA proteins depicted in the same way. Based on the current description it is not clear whether C-terminal a-helices of meiotic AAA+ proteins contact neighboring subunits differently (clockwise vs. counterclockwise) and whether they differ in their contributions to hexamer stabilization.

The authors should modify the figure caption to specify that the schematic is viewed from the C-terminus and should provide an additional schematic with the same colour code where the hexamer is viewed from the N-terminus.

3. Figures 3-5: It is unclear whether using SEM and the p-values (presumably calculated assuming normalcy) for PCC measurements is appropriate as PCC may not be normally distributed. A test of normalcy should be shown. Alternatively, a permutation test should be applied to calculate the p-values.

4. Page 14. Authors suggest that α11 couples subunit oligomerization with substrate binding, but the evidence presented suggests that ATAD1∆α11 simply fails to oligomerize robustly and that leads to reduced activity. If it does not oligomerize, it also won't bind peptide substrate. Therefore, inferring that α11 couples subunit oligomerization with substrate binding because peptide binding does not affect ATP hydrolysis seems weak. Can the authors provide more detail on the role of α11?

5. The authors speculate that a11 might bind to specific adaptors enabling targeting of ATAD1/Msp1 to the TOM complex or channeling substrates to the proteasome. Evidence for such role of a11 is however missing. It is also not clear how a11 should act as adaptor binding site. Its role in hexamer formation suggests that a11 accessibility will be largely restricted. This aspect should be taken into consideration.

6. Page 14/15. It is striking that the H206 residue is implicated so strongly in the structure with substrate binding, but has no effect in vivo. Why was the importance of H206 not analyzed for peptide binding in vitro to see if there was any defect using this more sensitive assay? The central substrate-binding pore-loop 1 of ATAD1 harbors two aromatic residues (W166/Y167). This is rather unusual as respective pore loops of other AAA+ proteins typically harbor only one aromatic residues. The authors propose that increased hydrophobicity of ATAD pore-loop 1 ensure a tighter grip on substrates, enabling ATAD1 to efficiently extract mislocalized TA-proteins from the lipid bilayer. While the authors demonstrate that Y167 can indeed not be replaced by a large aliphatic side chains, such an experiment was not performed for W166, which was only mutated to alanine. It is therefore crucial to also change W166 to valine or leucine residues to provide solid support for the original idea.

7. The authors state "By contrast, when expressed in the wild-type cells, ATAD1Da11 did not induce significant mislocalization of EGFP-Gos28 (Figure 4B), indicating its recessive phenotype, i.e., its lack of incorporation into and inactivating the wild-type hexamers present in these cells." This observation is very indirect, it could simply mean that incorporation is tolerated but maybe only up to a certain level of e.g. 2-3 subunits in the hexamer. In general, it would strengthen the work to run in vitro experiments where WT ATAD1 is substituted in solution with a11 deletion or mutants and the formation of hexameters as well as the ATPase activity is monitored.

8. The abstract is a bit confusing in that it mixes old and new data. The comparison between the old (Msp1) and new (ATAD1) structure should be organized more clearly.

*Reviewer #1 (Recommendations for the authors):*

Because of the conflicting results of the previous yeast assays suggesting that some structurally implicated aromatic pore 1 and 2 residues do not contribute to in vivo activity (Wang, et al., 2020) the authors develop a microscopy-based reporter assay (EGFP-Gos28) to monitor mistargeting of ATAD1 clients. They show that ATAD1∆a11 fails to remove reporters normally mistargeted to the mitochondria and does not show dominant effects. They demonstrate that both aromatic residues in pore 1 implicated for substrate binding in the current ATAD1 structure and in the prior Msp1 structure (Wang, et al., 2020) show deficiencies in removal of mistargeted proteins. Altering the histidine in pore 2 still has no effect on activity, calling into question the importance of this residue that is strongly implicated by the structure. Surprisingly, ATAD1 mutations implicated in neurological diseases do not seem to dramatically effect the function of ATAD1 in this assay.

Overall, this work extends our understanding of this family of AAA proteins responsible for extracting TA proteins mistargeted to the mitochondria. The role of the C-terminal α-helix 11 is clear from this work and the in vivo mislocalization reporter is a useful assay for sensitive in vivo activity of ATAD1 family proteins. It would be ideal to have additional in vivo assays (pull-downs, proximity labeling, etc) to validate these observations. Given the prior Msp1 structure from this same group and the largely similar observations, especially for the structural considerations, the conclusions of this work are not particularly surprising. Specific comments with the experimental methods, particularly with the microscopy assay, are detailed below.

Microscopy assay. Using mislocalization as a more sensitive assay is excellent. However, some of the statistics and reporting here could be improved. First, looking at the methods, it seems that the data shown is represented from a single experiment with multiple fields taken. Thus the mean PC shown is from a single biological replicate and therefore the variance/errors shown to compare between constructs is misleading. It would be ideal to have multiple biological replicates for each experimental condition and distinguish them from each other in the dot plots, representing the means for each of the replicates separately.

The other concern is that I'm not sure using SEM and the p-values (presumably calculated assuming normalcy) for PCC measurements as shown in figures 3-5 is appropriate as I don't know if PCC are normally distributed. It would be good to show test of normalcy or, better yet, use a permutation test to calculate the p-values. Ideally, promising results from this reporter assay would be confirmed with other assays in vivo, such as pull-downs, proximity labeling, or other evidence.

Page 14. The authors suggest that α11 couples subunit oligomerization with substrate binding, but all the evidence presented in this work suggests that ATAD1∆α11 simply fails to oligomerize robustly and that leads to reduced activity. If it does not oligomerize, it also won't bind peptide substrate. Therefore, inferring that α11 couples subunit oligomerization with substrate binding because peptide binding does not affect ATP hydrolysis seems weak.

Page 14/15. It is striking that the H206 residue is implicated so strongly in the structure with substrate binding, but has no effect in vivo. Was there a reason they did not test the importance of H206 in their in vitro peptide binding assays to see if there was any defect using this more sensitive assays?

*Reviewer #2 (Recommendations for the authors):*

The study reports on the cryo EM structure of ATAD1 and a novel activity assay that provides improved readout on ATAD1 derivatives in vivo. Both aspects of the study are nicely linked. The ATAD1 structure also illustrates new features (e.g. a11) as compared to published Ct Msp1. However, it remains largely unclear how those new features contribute to ATAD1 functional specificity (see comments below). This is reducing the novelty of the manuscript as the basic structural features have been described before (Ct Msp1). The manuscript is therefore in need of further experimental support and clarifications how unique sequence and structural features contribute to ATAD1/Msp1 specificity.

Figure 2: The representation of subunits is different from Figure 1. This leads to substantial confusion with respect to the relative order of individual subunits. While in Figure 1 M1 is the clockwise subunit with respect to M6, it becomes the counterclockwise one in Figure 2. The depiction in Figure 2 seems unusual. M1 represent the top subunit in the spiral arrangement of ATAD1 subunits and the seam subunit M6 is on its way to become the new top subunit (new M1) during subunit cycling. Therefore M1 is typically defined as counterclockwise subunit (see Figure 1) but not as the counterclockwise one (see Figure 2). The authors need to standardize the respective figures and data description and relate their findings to other AAA proteins depicted in the same way. Based on the current description it is not clear to this reviewer whether C-terminal a-helices of meiotic AAA+ proteins contact neighboring subunits differently (clockwise vs. counterclockwise) and whether they differ in their contributions to hexamer stabilization.

ATAD1/Msp1 differ from other meiotic AAA+ members by harboring an extended αa11 but not a short a11 that is followed by a12. For all AAA+ members the C-terminal a-helices contact the neighboring subunit and presumably contribute to hexamer stabilization and functional cycling. Why ATAD1/Msp1 harbor a long a11 remains unresolved. It is therefore unclear whether a11 contributes to functional specificity. The authors speculate that a11 might bind to specific adaptors enabling for ATAD1/Msp1 targeting to the TOM complex or channeling substrates to the proteasome. Evidence for such role of a11 is however missing. It is also not clear how a11 should act as adaptor binding site. Its role in hexamer formation suggests that a11 accessibility will be largely restricted.

The central substrate-binding pore-loop 1 of ATAD1 harbors two aromatic residues (W166/Y167). This is rather unusual as respective pore loops of other AAA+ proteins typically harbor only one aromatic residues. The authors propose that increased hydrophobicity of ATAD pore-loop 1 ensure a tighter grip to substrates, enabling ATAD1 to efficiently extract mislocalized TA-proteins from the lipid bilayer. While the authors demonstrate that Y167 can indeed not be replaced by a large aliphatic side chains, such experiment was not performed for W166, which was only mutated to alanine. It is therefore crucial to also change W166 to valine or leucine residues to provide solid support for the original idea.

*Reviewer #3 (Recommendations for the authors):*

There are a series of points I got curious about while reading the article. Maybe answers to these could make the paper even better.

1. Is there any indication from the structure on the mechanism by which ATAD1 facilitates re-insertion onto the right membrane?

2. In the Msp1 structure the authors describe the role of a0 as an important hook for substrate, what about the same element here? What about the lid domain in ATAD1?

3. In general the abstract could be changed, it is a bit confusing in mixing old and new data. It is actually good to compare with the Msp1 structure but then it has to be done in a clearer way e.g. on Msp1 we found this > on ATAD1 we find this.

4. Some references are not well stated, they only have ref written by the text.

5. To study better the motions of e.g. M6 in the closed state, did you try cryoDGRN or 3D variability in cryoSPARC?

6. The authors state "By contrast, when expressed in the wild-type cells, ATAD1Da11 did not induce significant mislocalization of EGFP-Gos28 (Figure 4B), indicating its recessive phenotype, i.e., its lack of incorporation into and inactivating the wild-type hexamers present in these cells." This observation is very indirect, it could simply mean that incorporation is acceptable but maybe only up to a certain level e.g. 2-3 subunits in the hexamer. In general, it would strengthen the work to run poisoning in vitro experiments where WT ATAD1 is substituted in solution with a11 deletion or mutants and the formation of hexameters as well as the ATPase activity is monitored.

7. When describing the density for the substrate in the map, you write that the side chains were clearly visible, but then you model a whole-alanine peptide stretch.

8. General question: can you say anything about the pulling force exerted by ATAD1 compared to other AAA+ of the same clade?

9. You state "ATAD1Y167F bound P13 with a ~10-fold weaker affinity than the wild-type enzyme, but still better than the other poreloop mutants. We reasoned that this difference resulted because the in vitro assay is more sensitive to small changes in activity that cell-based assays fail to capture". Could be instead because the cellular assay is only based on Gos28?

10. Discussing about the difference on a11 in ATAD1 or a11+a12in other AAA of the MC clade. Maybe a11 needs to be rigid? Have you considered inserting a G or P in the middle of a11 and see what happens?

11. In the table, also Relion should be mentioned, according to the material and methods.

12. The grids were prepared at a concentration of 100uM why so high? Was there a lot of aggregation? The SEC profile looks quite broad.

13. Looking at the actual maps. There is density present in the bottom of the closed structure at low threshold, what is it?

---

## [Author Response]

Reviewer #1 (Recommendations for the authors):Microscopy assay. Using mislocalization as a more sensitive assay is excellent. However, some of the statistics and reporting here could be improved. First, looking at the methods, it seems that the data shown is represented from a single experiment with multiple fields taken. Thus the mean PC shown is from a single biological replicate and therefore the variance/errors shown to compare between constructs is misleading. It would be ideal to have multiple biological replicates for each experimental condition and distinguish them from each other in the dot plots, representing the means for each of the replicates separately.

We agree that it is important to report on the number of biological replicates for each mutant. We have adjusted the figures with PCC measurements in two ways: 1) We have summarized the results from each imaging experiment in the corresponding supplemental figure. The mean value from each biological replicate is plotted for each mutant and the individual values are represented in the main text figures. 2) We have pooled the data from each biological replicate for the main text figures to better represent the data as a whole. The conclusions drawn from the experiments remain unaltered.

The other concern is that I'm not sure using SEM and the p-values (presumably calculated assuming normalcy) for PCC measurements as shown in figures 3-5 is appropriate as I don't know if PCC are normally distributed. It would be good to show test of normalcy or, better yet, use a permutation test to calculate the p-values. Ideally, promising results from this reporter assay would be confirmed with other assays in vivo, such as pull-downs, proximity labeling, or other evidence.

We have performed a normalcy test on the distributions for each mutant. For the most part, we do indeed see that they follow a normal distribution, however, there are a couple of mutants that do not. To account for this, we used the nonparametric Mann-Whitney test, which does not assume the normal distribution of data, to calculate the p-value. Each of the figures with PCC measurements have been updated to reflect this. The conclusions drawn from the experiments remain unaltered.

Page 14. The authors suggest that α11 couples subunit oligomerization with substrate binding, but all the evidence presented in this work suggests that ATAD1∆α11 simply fails to oligomerize robustly and that leads to reduced activity. If it does not oligomerize, it also won't bind peptide substrate. Therefore, inferring that α11 couples subunit oligomerization with substrate binding because peptide binding does not affect ATP hydrolysis seems weak.

We have modified our main text and indicated that instead of coupling substrate binding and oligomerization, the experiment simply shows that ATAD1∆a11’s ability to oligomerize is heavily impacted such that even the addition of a peptide substrate could not bring it to form functional oligomers.

Page 14/15. It is striking that the H206 residue is implicated so strongly in the structure with substrate binding, but has no effect in vivo. Was there a reason they did not test the importance of H206 in their in vitro peptide binding assays to see if there was any defect using this more sensitive assays?

Thanks for your suggestion. We purified the soluble ATAD1 protein bearing an H206A mutation. In our FP assay, we indeed detected a much weaker peptide binding affinity of this mutant compared to the wild-type protein. We incorporated the results in Figure 5 and also included them in the main text.

Reviewer #2 (Recommendations for the authors):Figure 2: The representation of subunits is different from Figure 1. This leads to substantial confusion with respect to the relative order of individual subunits. While in Figure 1 M1 is the clockwise subunit with respect to M6, it becomes the counterclockwise one in Figure 2. The depiction in Figure 2 seems unusual. M1 represent the top subunit in the spiral arrangement of ATAD1 subunits and the seam subunit M6 is on its way to become the new top subunit (new M1) during subunit cycling. Therefore M1 is typically defined as counterclockwise subunit (see Figure 1) but not as the counterclockwise one (see Figure 2). The authors need to standardize the respective figures and data description and relate their findings to other AAA proteins depicted in the same way. Based on the current description it is not clear to this reviewer whether C-terminal a-helices of meiotic AAA+ proteins contact neighboring subunits differently (clockwise vs. counterclockwise) and whether they differ in their contributions to hexamer stabilization.

Thank you for pointing out the different subunit depictions in Figures 1 and 2. In Figure 1, the membrane proximal side of ATAD1 is shown, as this is also the way that the previous Msp1 structures and other related AAA protein structures are displayed. In Figure 2, the opposite side of ATAD1, that is, the cytosol-facing side, is shown. The reason is because helix a11 resides on the cytosol-facing side, and we wanted to show the positions of a11. To avoid confusion, we added descriptions in the figure legends related to this point. Regarding the interactions made by a11, as shown in Figure 2 and Figure 2 —figure supplement 2, ATAD1 uses a11 to contact the counterclockwise subunit (looking from the cytosol-facing side) whereas other meiotic clade AAA proteins use a12 to contact the clockwise subunit.

ATAD1/Msp1 differ from other meiotic AAA+ members by harboring an extended αa11 but not a short a11 that is followed by a12. For all AAA+ members the C-terminal a-helices contact the neighboring subunit and presumably contribute to hexamer stabilization and functional cycling. Why ATAD1/Msp1 harbor a long a11 remains unresolved. It is therefore unclear whether a11 contributes to functional specificity. The authors speculate that a11 might bind to specific adaptors enabling for ATAD1/Msp1 targeting to the TOM complex or channeling substrates to the proteasome. Evidence for such role of a11 is however missing. It is also not clear how a11 should act as adaptor binding site. Its role in hexamer formation suggests that a11 accessibility will be largely restricted.

Based on the structure, a11 resides in the cytosol-facing side of the protein (see revised Figure 2). While its hydrophobic side is buried, its hydrophilic side is exposed to the cytosol (Figure 2E), available for potential adaptor binding interactions. We agree with the reviewer that the precise reason why ATAD1 has a long a11 instead of a short a11 and a short a12 is not yet clear, thus we only mentioned this adaptor-binding function as a plausible possibility in the discussion.

The central substrate-binding pore-loop 1 of ATAD1 harbors two aromatic residues (W166/Y167). This is rather unusual as respective pore loops of other AAA+ proteins typically harbor only one aromatic residues. The authors propose that increased hydrophobicity of ATAD pore-loop 1 ensure a tighter grip to substrates, enabling ATAD1 to efficiently extract mislocalized TA-proteins from the lipid bilayer. While the authors demonstrate that Y167 can indeed not be replaced by a large aliphatic side chains, such experiment was not performed for W166, which was only mutated to alanine. It is therefore crucial to also change W166 to valine or leucine residues to provide solid support for the original idea.

Thank you for the suggestion. We made the W166V and the W166L mutations in ATAD1 and tested the mutants’ function in both the cell-based assay and the peptide binding assay. As shown in the updated Figure 5, both ATAD1^W166V^ and ATAD1^W166L^ showed reduced activity in extracting mislocalized Gos28 in cells. In agreement with this result, both mutants showed reduced peptide binding activity in the peptide binding assay (see updated Figure 5). Together, these results show that ATAD1 requires both aromatic amino acids (W166 and Y167) in pore-loop 1 for its proper function.

Reviewer #3 (Recommendations for the authors):There are a series of points I got curious about while reading the article. Maybe answers to these could make the paper even better.1. Is there any indication from the structure on the mechanism by which ATAD1 facilitates re-insertion onto the right membrane?

This is a good question. The current structure only captures the substrate bound in the central pore. However, previous studies (Matsumoto et al., Mol Cell 2019; Dederer et al., *eLife* 2019) on Msp1 showed that some extracted tail-anchored proteins are re-inserted to the ER membrane where they are either degraded or trafficked from there to the correct organelle. This has been shown for the yeast Msp1, but not yet for the mammalian ATAD1. It also remains unclear what factors are needed to facilitate the translocation from the mitochondria to the ER, whether is it a direct handover by Msp1/ATAD1, or requires another chaperone/targeting factor is unknown.

2. In the Msp1 structure the authors describe the role of a0 as an important hook for substrate, what about the same element here? What about the lid domain in ATAD1?

Thank you for raising this point. Many amino acids that are involved in substrate binding in Msp1’s N-domain are conserved in ATAD1 as well (Li et al., EMBO J 2019). Thus, it is possible that they are also important in engaging substrates in ATAD1. In the Msp1 structure, we observed the conformational change in helix a0 within the hexamer. We proposed that the melting of this helix reveal the hydrophobic amino acids that could constitute a potential substrate binding site (Wang et al., 2020 *eLife*). In the ATAD1 structures, we observed similar conformational changes in helix a0 (see Author response image 1), and similar positioning of the hydrophobic amino acids within the linker domain.

**Author response image 1. sa2fig1:** Structure details of the linker domain (LD). (**A**) EM map of D40-ATAD1 (open) showing that helix a0 within the LD is well-folded in the central four subunits and disordered in M1 and M6. (**B**) and (**C**) showing most of the hydrophobic amino acids within the LD are buried by the well-folded a0 and revealed when a0 becomes disordered. These hydrophobic amino acids make up a plausible substrate binding site. a0 is colored in white and hydrophobic amino acids in gold. Helix a0 is colored in green, hydrophobic amino acids in gold, and the substrate peptide is colored in black.

3. In general the abstract could be changed, it is a bit confusing in mixing old and new data. It is actually good to compare with the Msp1 structure but then it has to be done in a clearer way e.g. on Msp1 we found this > on ATAD1 we find this.

Thank you for your suggestion. We modified our abstract to more clearly distinguish the new/ATAD1-specific insights from what was previously known from Msp1.

4. Some references are not well stated, they only have ref written by the text.

Thank you for spotting this. We fixed the references.

5. To study better the motions of e.g. M6 in the closed state, did you try cryoDGRN or 3D variability in cryoSPARC?

Yes, we tried the 3D variability analysis in cryoSPARC. One of the modes of motion showed clearly that the M6 subunit is indeed mobile and moves around the equilibrium position both vertically and horizontally. Please see Author response image 2 for different states captured by the 3D variability analysis.

**Author response image 2. sa2fig2:** 3D variability analysis of D 40-ATAD1^E193Q^Da11 (open) 3D variability analysis was performed in cryoSPARC to capture the movements of the flexible regions of the molecule. Twenty different frames were used to capture the full trajectory of the movements. Within the twenty frames, the start and the end frames (representing the two extreme states of the movement) are overlayed and shown in the figure. The M6 subunit is colored in blue (start frame) or pink (end frame) and the rest of the protein in light grey.

6. The authors state "By contrast, when expressed in the wild-type cells, ATAD1Da11 did not induce significant mislocalization of EGFP-Gos28 (Figure 4B), indicating its recessive phenotype, i.e., its lack of incorporation into and inactivating the wild-type hexamers present in these cells." This observation is very indirect, it could simply mean that incorporation is acceptable but maybe only up to a certain level e.g. 2-3 subunits in the hexamer. In general, it would strengthen the work to run poisoning in vitro experiments where WT ATAD1 is substituted in solution with a11 deletion or mutants and the formation of hexameters as well as the ATPase activity is monitored.

To clarity the question, our interpretation of the data is that ATAD1 Da11 failed to poison the wild-type hexamers, because its ability to oligomerize with wild-type ATAD1 is impacted. The alternative interpretation raised by the reviewer is that perhaps the intrinsic ability for ATAD1Da11 to oligomerize with wild-type subunits is not impacted, but rather, in our experiments, due to low expression levels, only 2-3 subunits of ATAD1Da11 got incorporated into the each WT hexamer, and therefore it could not poison the WT hexamer. We do not think this is the case, we compared our observations of ATAD1Da11 to those of ATAD1^E193Q^. First, both two mutants are inactive, that is, in ATAD1 knockout cells, neither could remove mislocalized Gos28 from the mitochondria (see Figure 4A in the manuscript). However, in wildtype cells, one mutant (ATAD1^E193Q^) managed to completely poison the wild-type enzyme (dominant negative), and the other (ATAD1Da11) had little effect (see Figure 4B in the manuscript). Interestingly, both mutants were expressed at similar levels in wild-type cells (see Author response image 3 for quantification). This is to say that at similar expression levels, one inactive mutant (ATAD1^E193Q^) could poison the wild-type enzyme, and the other inactive mutant (ATAD1Da11) could not. Having ruled out the expression level difference, the most plausible explanation to this phenotypic difference between the two mutants is that, ATAD1^E193Q^ can oligomerize with wild-type ATAD1 efficiently (as we know this is a property of the “Walker B” mutant), but ATAD1Da11 has an impaired ability to oligomerize, therefore was not incorporated into a hexamer formed mostly of wild-type subunits.

**Author response image 3. sa2fig3:** Quantification of ATAD1 variant expression levels in the WT background. The average fluorescent intensity for cells expressing ATAD1-HaloTag were plotted for each biological replicate as a proxy for expression levels. ATAD1Da11 is expressed at similar, if not higher, levels when compared to ATAD1^E193Q^.

7. When describing the density for the substrate in the map, you write that the side chains were clearly visible, but then you model a whole-alanine peptide stretch.

The side chain density is indeed clearly visible as shown in Figure 1, meaning that there is density that goes beyond the peptide backbone. However, we could not assign the identity of the side chains based on the current density, which is why a poly-alanine stretch was modeled.

8. General question: can you say anything about the pulling force exerted by ATAD1 compared to other AAA+ of the same clade?

To compare the force exerted by ATAD1 and other AAA proteins, specific force measurements will have to be performed using a single molecule setup that go beyond the scope of this paper. However, we did comment on this point in our discussion, speculating that the extra aromatic amino acids possibly give ATAD1 a stronger grip, which adapts it for its function of removing membrane proteins.

9. You state "ATAD1Y167F bound P13 with a ~10-fold weaker affinity than the wild-type enzyme, but still better than the other poreloop mutants. We reasoned that this difference resulted because the in vitro assay is more sensitive to small changes in activity that cell-based assays fail to capture". Could be instead because the cellular assay is only based on Gos28?

Yes, that is also a possibility that the differences between the in vitro and the cellbased assays arise from the different substrates used. We included this possibility in our revised manuscript.

10. Discussing about the difference on a11 in ATAD1 or a11+a12in other AAA of the MC clade. Maybe a11 needs to be rigid? Have you considered inserting a G or P in the middle of a11 and see what happens?

We have indeed considered this, however, trying to break a11 into two shorter helices that mimic the a11+ a12 arrangement in other AAA proteins has one problem, which is the a12 region would have to be adapted to interacting with the counterclockwise subunit. Thus, if the experiment shows that breaking a11 kills ATAD1’s activity, it could either mean that the a11+ a12 arrangement does not work for ATAD1, or simply the new “a12” is not properly positioned to interact with the rest of the protein. Also, given that a11 is an amphipathic helix (see Figure 2E), inserting one amino acid in between will cause shifts in the positions of the side chains which will alter the way the helix interacts with the rest of the protein. Thus, we do not think this is a good experiment to study the role of a11.

11. In the table, also Relion should be mentioned, according to the material and methods.

Thanks for pointing this out. We added Relion in Table 1.

12. The grids were prepared at a concentration of 100uM why so high? Was there a lot of aggregation? The SEC profile looks quite broad.

We added some detergent to help improve particle orientation distribution (same protocol as used in the previous Msp1 purification). The addition of detergent drives particles towards carbon, which is why a higher protein concentration was required to see the same concentration of particles in ice. The majority of particles were well-dispersed in most micrographs and we were able to obtain high-resolution reconstructions based on those particles. The SEC profile was indeed broad, as is also seen in the previous Msp1 purification (Wang et al., 2020 *eLife*). This is not surprising as we have observed that both Msp1 and ATAD1’s oligomerization is dependent on the subunit concentration, thus some on-column dissociation of hexamers might have happened that causes the broad peaks.

13. Looking at the actual maps. There is density present in the bottom of the closed structure at low threshold, what is it?

Thank you for noticing this. That is most likely the 6XHis-tag at the N-terminus of the protein that was not cleaved off.